

# Eddy enhanced primary production accelerates bacterial growth in the
# Eastern Tropical North Atlantic
Quentin Devresse[1], Kevin W. Becker[1], Arne Bendinger[1,2], Johannes Hahn[1,3], Anja Engel[1]
[1]GEOMARHelmholtz Centre for Ocean Research Kiel, Germany,
[2] Laboratoire d'Etudes en Géophysique et Océanographie Spatiales (LEGOS), Université Toulouse,
IRD, CNRS, CNES, UPS, Toulouse, France
[3] Bundesamt für Seeschifffahrt und Hydrographie, Hamburg, Germany
Correspondence: Quentin Devresse (qdevresse@geomar.de)
## Abstract
Mesoscale eddies play essential roles in modulating the ocean's physical, chemical, and
biological properties. In cyclonic eddies (CE) nutrient upwelling can stimulate primary
production by phytoplankton. Yet, how this locally enhanced autotrophic production affects
heterotrophic bacterial activities (biomass production and respiration) and consequently the
metabolic balance between the synthesis and the consumption of dissolved organic matter
(DOM) remains largely unknown. To address this gap, we investigated the horizontal and
vertical variability of phytoplankton and heterotrophic bacterial activity along ~900 km zonal
corridor between the coast of Mauretania and the Cape Verde Islands in the eastern tropical
North Atlantic (ETNA). We additionally collected samples from a CE along this transect at
high spatial resolution. Our results show cascading effects of physical disturbances induced by
a CE on phyto- and bacterioplankton biomass and metabolic activities. Specifically, the
injection of nutrients into the sunlit surface resulted in enhanced autotrophic plankton
abundance and activity as indicated by Chlorophyll *a* (Chl-*a*) concentration, DOM exudation,
and primary productivity (PP). However, the detailed eddy survey revealed an uneven
distribution of these parameters with, for example, the highest Chl-*a* concentrations and PP
rates near and just beyond the CE's periphery. The heterotrophic bacterial activity was similarly
variable. Optode-based bacterial respiration (BR) and biomass production (BP) largely
followed the trends of PP and Chl-*a*. Thus, a submesoscale spatial mosaic of heterotrophic
bacterial abundance and activities occurred within the CE studied here that was closely related
to variability in autotrophic production. This was supported by a significant positive correlation



between concentrations of semi-labile organic carbon (SL-DOC; the sum of dissolved
hydrolyzable amino acids and combined carbohydrates) and BR measurements. Bacterial
growth efficiency (BP/( BR+BP)) was variable (1.4-10.5%) within the CE and carbon
exudation was not always sufficient to compensate the bacterial carbon demand (BR+BP; 28.3-
114.5%). We have additionally estimated the metabolic state in our samples, which showed that
the CE carried a strong autotrophic signal (PP/(BR+BP)>1). Overall, our results show that
submesoscale (0-10 km) processes lead to highly variable metabolic activities of both
phototrophic and heterotrophic microbes, which has implications for biogeochemical models
estimating oceanic carbon fluxes. Additionally, we revealed that the CE not only traps and
transports coastal nutrients and carbon to the open ocean but also stimulates phytoplankton
growth generating freshly produced organic matter during westward propagation. This organic
matter may fuel heterotrophic processes in the open ocean and may help to explain the often-
observed net heterotrophic metabolic state of these environments.

## 1. Introduction


Mesoscale eddies (10-100 km) are ubiquitous in the ocean affecting upper ocean
biogeochemistry and ecology, e.g. upwelling nutrients influencing primary production and
carbon export (Cheney and Richardson, 1976; Arístegui et al., 1997). The sense of rotation and
their vertical structure classifies cyclonic (CEs), anticyclonic (ACEs; e.g. Chelton et al., 2011)
or anticyclonic mode water eddies (ACMEs; D'Asaro 1988). In Eastern Boundary Upwelling
Systems (EBUS), eddies may form by flow separation of along slope boundary currents at
topographic headlands (D'Asaro 1988, Molemaker et al., 2015, Thomsen et al., 2016). Eddies
have lifespans from days to months and can travel several hundred to thousands of kilometers
across ocean basins (Chelton et al., 2011). They are complex dynamical regimes for organic
matter and nutrient transport (Gruber et al., 2011). In the North Atlantic Ocean, eddies
generated in the highly productive Canary Upwelling System (CanUS) may laterally propagate
to the oligotrophic Subtropical North Atlantic Gyre (SNAG), transporting thereby nutrients and
carbon (McGillicuddy et al., 2003; Karstensen et al., 2015; Schütte et al., 2016). A variety of
studies demonstrated the impact of eddies on primary production (PP) on a global scale. Yet,
the magnitude of the eddy-induced flux and its utilization depend on the model, the area
investigated, and the degree of resolution and is still controversial (See review by
McGillicuddy, 2016 and references therein). For example, Couespel et al., (2021) performed



global warming simulations using a representation of mid-latitude double-gyre circulation and
showed that at the finest model resolution (1/27°), eddies can mitigate the decline of primary
production (−12 % at 1/27° vs. −26 % at 1°). Modeling studies have long urged consideration
of the effects of eddies on PP at submesoscale levels (0.1-10 km) to provide realistic estimates
of the oceanic carbon cycle (Levy et al., 2001). Thus, understanding the impact of mesoscale
eddies on plankton productivity will help to better predict future carbon cycling in EBUS under
global change scenarios.
Eddies modulate the mixed layer depth by upwelling (CEs), downwelling (ACEs), or
frontogenesis from eddy-eddy interaction, thereby creating spatial variability of nutrient
concentration within/around eddies on length scales of 0.1-10 km (see reviews by Mahadevan,
2016 and McGillicuddy, 2016). In addition, the nonlinear response of phytoplankton growth to
nutrient availability and advection of phytoplankton by currents makes plankton distribution
and community composition highly variable within and around eddies (Lochte and Pfannkuche
1987). As a consequence, the spatial distribution of PP across eddies can be highly variable
(e.g. Falkowski et al., 1991; Ewart et al., 2008; Singh et al., 2015). Still, insight into the
distribution of phytoplankton and their activities within mesoscale eddies is limited due to a
lack of sufficient fine-scale vertical and horizontal resolution studies to adequately describe
these distributions.
Bacterial activity is directly coupled to PP: autotrophic cells release dissolved organic matter
(DOM), the main substrate for heterotrophic bacteria and archaea (Thornton 2014). DOM
release has been interpreted as a cellular overflow mechanism that expels the carbon produced
in excess (Wood and Van Valen, 1990; Schartau et al., 2007). Therefore, released DOM
compounds are often depleted in nutrients limiting autotrophic cell growth (Engel et al., 2002).
Patchiness of phytoplankton primary productivity and nutrient limitation within eddies may
thus lead to spatial heterogeneity of extracellular release rates (e.g. Lasternas et al., 2013, Rao
et al., 2021) with distinct quality (e.g. Wear et al., 2020). DOM quality impacts biomass
production (BP), bacterial respiration (BR), and, thus the bacterial growth efficiency (BGE;
Neijssel and de Mattos, 1994; Russell and Cook, 1995). BGE is the ratio between BP and the
bacterial carbon demand (BCD), which is the sum of assimilated carbon that is respired and
carbon that is incorporated into biomass (BP + BR). Lønborg et al., (2011) established that BGE
decreases with increasing C/N ratio of the bioavailable DOM produced by phytoplankton. BGE
is a critical parameter for estimating the amount of consumed organic carbon that is used to
build biomass by heterotrophic bacteria (Anderson and Ducklow 2001). So far, BGE within



eddies has been reported for ACEs from the Mediterranean Sea (Christaki et al., 2011), but not
for CEs and Mode Water Eddies. In general, several studies showed a patchy distribution of
bacterial abundance, BP (Ewart et al., 2008; Baltar et al., 2010), BR, community respiration
(CR) (Mouriño-Carballido and McGillicuddy 2006; Mouriño-Carballido, 2009), and of the
metabolic balance between production and consumption of organic matter (Maixandeau et al.,
2005; Ewart et al., 2008; Mouriño-Carballido and McGillicuddy 2006; Mouriño-Carballido,
2009) within eddies.
Yet, how eddies affect microbial plankton dynamics and carbon flow is largely unknown. So
far, phyto- and bacterioplankton distribution and activities were either studied separately or at
relatively low spatial resolution. Data on eddy-induced changes in primary production,
extracellular release and semi-labile DOM concentration, and the responses of heterotrophic
microbial metabolic activities are scarce. Understanding how eddies modulate microbial
activities will enhance our knowledge about the fate of autotrophically fixed organic carbon
and the overall $CO_2$ source/sink function in the ocean, and in particular EBUS.
Here, we studied the impact of a CE on microbial carbon cycling along a zonal corridor of the
westward propagating eddies between the Cape Verde Islands and the Mauretania Upwelling
System 13-20 °N), a sub-region of the CanUS (13-33 °N, Arístegui et al., 2009). About 146 ± 44
eddies with a lifetime of more than 7 days are generated per year in this region (Schütte et al.,
2016). Along this corridor, we determined phytoplankton (<20µm) cell abundance, primary
production, and extracellular release. We linked those parameters of autotrophic activity to
semi-labile DOM concentration and heterotrophic bacterial activity. Our study gives new
insights into 1) microbial carbon cycling and 2) factors controlling microbial metabolic
activities within and around CE formed in EBUS.

## 2. Materials and Methods


### 2.1 Study area and eddy characterization


Sampling was conducted in the ETNA between the Cape Verde archipelago and the
Mauritanian coast during cruise M156 (July 3rd to August 1st, 2019. Figure **1A**) on the R/V
*Meteor*. Samples were collected during the relaxation period (from May to July) that follows
the upwelling season (January to March; Lathuilière et al., 2008). A CE was sampled at high





spatial resolution along two zonal (from 19.1 °W to 18.2 °W at 18.3 °N and from 18.5 °W to
17.1 °W at 18.6 °N) and one meridional transects (from 19.4 °N to 18 °N at 18.4 °W to 18.1
°W). The zonal section was slightly meridionally shifted east/west of the eddy core position.
The reason for that was the deformed eddy shape, which resulted in a consecutive optimized
identification of the eddy core position during the eddy survey. In addition, we sampled water
along the 18 °N transect, a typical coast to open ocean trajectory of eddies in the region (Schütte
et al., 2016). Salinity, temperature, depth, and $O_2$ concentration were determined at each station
using a Seabird 911 plus CTD system equipped with two independently working sets of
temperature-conductivity-oxygen sensors. The oxygen sensor was calibrated against discrete
water samples using the Winkler method (Strickland and Parsons, 1968; Wilhelm, 1888).
Seawater samples were collected from the top 200 m using 10L Niskin bottles attached to the
CTD Rosette. A total of 25 stations were sampled; 14 of them inside or in the vicinity of the
CE. Sampling was conducted in the epipelagic layer (0-200 m), including water from the
surface, within the mixed layer, at the Chl-*a* maximum, and within the shallow oxygen
minimum zone (OMZ; <50 µmol kg$^{-1}$ between 0-200 m depth) when present.
Sea surface height (SSH) and Acoustic Doppler Current Profiler (ADCP) velocity data (SI Fig.
**1**), characterized the eddy as a CE. Based on the Angular Momentum Eddy Detection and
Tracking Algorithm (AMEDA; Le Vu et al., 2018), the eddy was estimated to be 1.5 months
old. The center of the eddy and the core radius were determined using ADCP reconstruction
assuming an axis-symmetric vortex. (SI Fig. **1**). On 22/07/2019, the eddy center was located at
18.69 °N, 18.05 °W, with a core radius of 40.5 ± 5.7 km. The mean azimuthal velocity in the
CE was 19.9 ± 0.7 cm s$^{-1}$ and the absolute dynamic topography associated with the CE core
was ~23 cm on 23/07/19. Fine-scale analysis of the eddy physics will be given by Fischer et al.
(2022, in prep). However, as the eddy shape was deformed, ADCP reconstruction did not
constrain well the physical border of the eddy (SI Fig. **1**).  Therefore, we combined sea surface
temperature (23.44 ± 0.47 °C) salinity (39.95 ± 0.04) and Chl-*a* (1.35 ± 0.73 µg L$^{-1}$) data to
approximate the area influenced by the eddy (Fig. **1b,c,d**). We classified stations into 'core'
and 'periphery' of the eddy. Stations that were outside and westward of the eddy influence were
referred to as 'open ocean' and those close to the coast as 'coastal'. At the St. E3, outside of the
CE periphery, we observed a front with surface temperature and salinity (not compensating in
density) being clearly different from among the adjacent stations (Fig. **1b**), potentially which
might be related to enhanced, an up- and downwelling might have occurred there on either side
of the front, respectively. Hence, we referred to that station as 'Frontal Zone'. The classification




of stations is thoroughly discussed in the supplementary information (SI), and the sampling
time, location, and distance from the eddy center are given in Table S**1**.

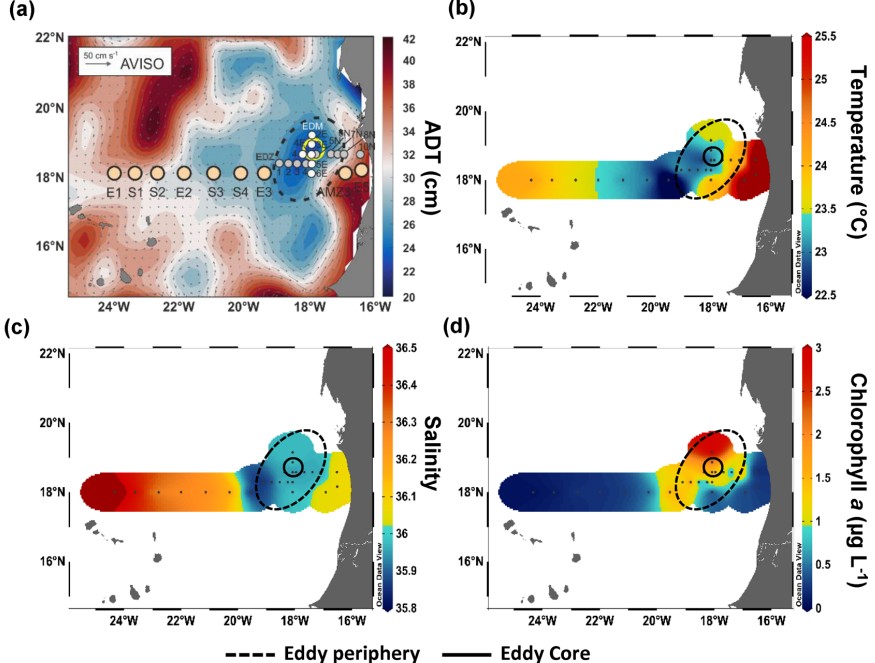


Figure 1: M156 cruise track (**a**) Temperature at 5m depth (**b**) Salinity at 5m depth (**c**) chlorophyll a at
5m depth (**d**). The color background in (**a**) shows the variations in Absolute Dynamic Topography
(ADT). The direction and speed of surface water geostrophic currents are shown as arrows.

2.2 Chemical analyses
Nutrient concentrations were determined at selected stations (SI Table **1)**. Nutrients were
measured onboard from duplicate samples (11 mL) of unfiltered seawater samples. Ammonium
($NH_4^+$) was analyzed after Solórzano (1969) and phosphate ($PO_4$), nitrate ($NO_3$), nitrite ($NO_2$),
and silicate ($Si(OH)_4$) were measured photometrically with continuous-flow analysis on an
auto-analyzer (QuAAtro; Seal Analytical) after Grasshoff et al., (1999). Detection limits for
$NH_4^+$, $PO_4$, $NO_3$, $NO_2$, and $Si(OH)_4$ were 0.1, 0.02, 0.1, 0.02, and 0.2 µmol $L^{-1}$, respectively
Total dissolved inorganic nitrogen (DIN) was determined as the sum of $NH_4^+$, $NO_3$, and $NO_2$.



To estimate the fraction of semi-labile dissolved organic carbon (DOC), we determined high-molecular-weight (HMW> 1 kDa) dissolved combined carbohydrates (dCCHO) and dissolved amino acids (dAA) as the main biochemical components of DOM.

Duplicate samples (20 mL) for dCCHO were filtered through 0.45 µm Acrodisk filters, collected in combusted glass vials (8 h, 450 °C) and frozen (−20 °C) until analysis after Engel & Händel (2011) with a detection limit of 1 µg L$^{-1}$. The analysis detected 11 monomers: arabinose, fucose, galactose, galactosamine, galacturonic acid, glucosamine, glucose, glucuronic acid, rhamnose, co-elute mannose, and xylose.

Duplicate samples (4 mL) for dHAA were filtered through 0.45µm Acrodisk filters, collected in combusted glass vials (8 h, 450 °C), and frozen (−20 °C) until analysis. dAA were measured with ortho- phthaldialdehyde derivatization by high-performance liquid chromatography (HPLC; Agilent Technologies, USA) equipped with a $C_{18}$ column (Phenomenex, USA) (Lindroth and Mopper, 1979; Dittmar et al., 2009). The analysis classified 13 monomers with a precision < 5 % and a detection limit of 2 nmol L$^{-1}$: alanine, arginine, aspartic acid, isoleucine, glutamic acid, glycine, leucine, phenylalanine, serine, threonine, tyrosine, valine; and γ-aminobutyric acid (GABA).

The calculations for the carbon content of dCCHO and dHAA were based on carbon atoms contained in the identified monomers. The sum of dCCHO and dHAA carbon content is referred to as semi-labile DOC (SL-DOC).

For Chl-*a,* 1L samples were collected on 25 mm GF/F (Whatman, GE Healthcare Life Sciences, UK) and subsequently frozen (−20 °C) until extraction using 90 % acetone for photometric analyses (Turner Designs, USA), slightly modified after Evans et al., (1987).

Bacteria were quantified using a flow cytometer (FACSCalibur, Becton Dickinson, Oxford, UK). Seawater samples (1.7 mL) were fixed with 85 µL glutaraldehyde (1% final concentration) and stored at -80 °C until enumeration. Samples were stained with SYBR Green I (molecular probes) and were enumerated with a laser emitting at 488 nm and detected by their signature in a plot of side scatter (SSC) vs green fluorescence (FL1). Heterotrophic bacteria were distinguished from photosynthetic bacteria (*Prochlorococcus* and *Synechococcus*) by their signature in a plot of red fluorescence (FL2) vs green fluorescence (FL1). Yellow-green latex beads (1 µm, Polysciences) were used as an internal standard. (Stolle et al., 2009). Cell counts were determined with the CellQuest software (Becton Dickinson). For autotrophic pico and nanoplankton <20 µm, 2 mL samples were fixed with formaldehyde (1 % final concentration)





and stored frozen (−80 °C) until analysis. Red and orange autofluorescence was used to identify
Chl-*a* and phycoerythrin cells. Cell counts were determined with CellQuest software (Becton
Dickinson); picoplankton and nanoplankton populations containing Chl-*a* and/or phycoerythrin
(i.e., *Synechococcus*) were identified and enumerated. We converted the cell abundance of the
different autotrophic plankton populations into biomass assuming 43 fg C cell$^{-1}$ for
*Prochlorococcus*, 120 fg C cell$^{-1}$ for *Synechococcus*, 500 fg C cell$^{-1}$ for eukaryotic picoplankton
and, 3.100 fg C cell$^{-1}$ for eukaryotic nanoplankton after Hernández-Hernández et al., (2020).
We report the autotrophic plankton biomass as the sum of eukaryotic pico- and nanoplankton
and cyanobacteria (*Prochlorococcus* and *Synechococcus*) biomass. The abundance of
eukaryotic pico- and nanoplankton and cyanobacteria (*Prochlorococcus* and *Synechococcus*)
can be found in the SI (Table S**2**).

2.3 Microbial activities
More information on procedures and calculations of microbial activities are given in the SI.
Bacterial biomass production rates (BP) were measured through the incorporation of labeled
leucine ($^3$H) (specific activity 100 Ci mmol$^{-1}$, Biotrend) using the microcentrifuge method
(Kirchman et al., 1985; Smith and Azam, 1992). Duplicate samples and one killed control (1.5
mL each) were labeled using $^3$H-leucine at a final concentration of 20 nmol L$^{-1}$ and incubated
with headspace for 6 h in the dark at 14 °C. Controls were poisoned with trichloroacetic acid.
All Samples were measured on board with a liquid scintillation analyzer (Packard Tri-Carb,
model 1900 A). $^3$H-leucine uptake was converted to carbon units applying a conversion factor
of 1.55 kg C mol$^{-1}$ leucine (Simon and Azam, 1989).
BP rates at 22 °C were estimated following López-Urrutia and Morán (2007):

$$BP_{22°C} = BP_{14°C} \times 0.996 \quad (Eq. 1)$$

*C*ommunity respiration rates (CR) were estimated from changes of dissolved oxygen in 24-36
hours incubations at 14°C using optode spot mini sensors (PreSens PSt3; Precision Sensing
GmbH, Regensburg, Germany). The detection limit (DL) for CR was 0.55 µmol O$_2$ L$^{-1}$ d$^{-1}$.
CR at 22°C was estimated using extrapolation from Regaudie-De-Gioux and Duarte (2012):

$$CR_{22°C} = CR_{14°C} \times 2.011 - 0.013 \quad (Eq. 2)$$

CR$_{22°C}$ was converted into bacterial respiration (BR$_{22°C}$) after Aranguren-Gassis et al. (2012):



$$BR_{22°C} = 0.30 \times CR_{22°C}^{1.22} - 0.013 \qquad (Eq.\ 3)$$
A respiratory quotient of 1 was used to convert oxygen consumption into carbon respiration
(del Giorgio and Cole 1998).
We furthermore estimated the bacterial carbon demand (BCD):
$$BCD = BP + BR \qquad (Eq.\ 4)$$
and the bacterial growth efficiency (BGE):
$$BGE = \frac{BP}{BCD} \qquad (Eq.\ 5)$$
Primary production (PP) was determined from $^{14}$C incorporation according to Steemann
Nielsen (1952) and Gargas (1975). Polycarbonate bottles (Nunc EasYFlask, 75 cm$^2$) were filled
with 260 mL prefiltered (mesh size of 200 μm) sample and spiked with 50 μL of a ~11 μCi
NaH$^{14}$CO$_3^-$ solution (Perkin Elmer, Norway). 200 μL were removed immediately after spiking
and transferred to a 5 mL scintillation vial for determination of added activity. Then, 50 μL of
2N NaOH and 4 mL scintillation cocktail (Ultima Gold AB) were added. Duplicate samples
were incubated in 12 h light and 12 h dark at 22 °C. Three light levels were applied: 1200-1400;
350 and 5 μE, with high values representing surface irradiance at the time of sampling. The
incubation length was chosen for two reasons. First, we expected low productivity of the open
ocean phytoplankton community due to low biomass and low nutrient concentrations at the start
of the incubation. Under these conditions, short-term incubations of only a few hours may
underestimate PP, because carbon assimilation by algal cells may be too low to discriminate
against $^{14}$C adsorption as determined in blank dark incubation (Engel et al., 2013). Moreover,
the release of freshly assimilated carbon into the DOM pool has a time scale of several hours
because of the equilibration of the tracer and because metabolic processes of organic carbon
exudation follow those of carbon fixation inside the cell (Engel et al., 2013). Incubations were
stopped by filtration of a 70 mL sub-sample onto 0.4 μm polycarbonate filters (Nuclepore).
Particulate primary production (PP$_{POC}$) was determined from material collected on the filter,
while the filtrate was used to determine dissolved primary production (PP$_{DOC}$). All filters were
rinsed with 10 mL sterile filtered (<0.2 μm) seawater, and then acidified with 250 μL 2N HCl
to remove inorganic carbon (Descy et al., 2002). Filters were transferred into 5 mL scintillation
vials, and 4 mL scintillation cocktail (Ultima Gold AB) was added. To determine PP$_{POC}$ and
PP$_{DOC}$, 4 mL of filtrate and incubated sample were transferred to 20 mL scintillation vials,
acidified (100 μL 1N HCl), and left open in the fume hood to remove inorganic carbon. Then,



100 μL of 2N NaOH and 15 mL scintillation cocktail were added. All samples were counted
the following day in a liquid scintillation analyzer (Packard Tri-Carb, model 1900 A).
Primary production (PP) of organic carbon was calculated according to Gargas (1975):

$$\text{PP } (\mu\text{molC L}^{-1}\text{ d}^{-1}) = \frac{a2 \times DI^{12}C \times 1.05 \times k_1 \times k_2}{a1} \quad \text{(Eq.6)}$$

Where $a1$ and $a2$ are the activities (DPM) (disintegrations per minute) of the added solution
and the sample corrected for dark sample, respectively, and DI$^{12}$C is the concentration (μmol
L$^{-1}$) of dissolved inorganic carbon (DIC) in the sample. Dissolved inorganic carbon
concentration was calculated from total alkalinity using r package seacarb (Gattuso et al., 2020).
Total alkalinity of the seawater was acquired through the open-cell titration method (Dickson
et al., 2007). The value 1.05 is a correction factor for the discrimination between $^{12}$C and $^{14}$C,
as the uptake of the $^{14}$C isotope is 5% slower than the uptake of $^{12}$C, $k_1$ is a correction factor
for subsampling (bottle volume/filtered volume) and $k_2$ is the incubation time (d$^{-1}$). Total
primary production (PP$_{TOT}$; μmol C L$^{-1}$ d$^{-1}$) was derived from the sum of PP$_{POC}$ and PP$_{DOC}$
according to:

$$\text{PP}_{TOT} = \text{PP}_{POC} + \text{PP}_{DOC} \quad \text{(Eq.7)}$$

The percentage of extracellular release (PER; %) was calculated as:
$$\text{PER} = \left(\frac{\text{PP}_{DOC}}{\text{PP}_{TOT}}\right) \times 100 \quad \text{(Eq.8)}$$

2.4 Data analysis
Statistical analyses and calculations were conducted using the software R (v4.0.3) in Rstudio
(v1.1.414; Ihaka and Gentleman 1996). Analysis of variances (ANOVA) and Tukey test, were
performed on the different parameters by grouping the station by their position (SI Table **1**).
Seawater density was calculated using r package oce v1.3.0 (Kelley, 2018) and mixed layer
maximum depth was determined as the depth at which a change from the surface density of
0.125 has occurred (Levitus, 1982). Section plots were realized using Ocean Data View
(Schlitzer, 2020). Other packages used in this study include corrplot v0.84 (Dray, 2008) and
ggplot2 v3.3.3 (Wickham, 2016). Depth integrated values were calculated using the midpoint
rule.



## 3. Results

### 3.1 Hydrographic conditions

Along the zonal transect, open ocean waters (from 20 to 24.5 °W) had a temperature range of 17.0-24.3 °C and salinity of 36.19-36.79 in the upper 150m depth (Fig. **2a** & **b**). The average mixed layer depth was $30 \pm 2$ m (SI Table **1**). Oxygen concentration (Fig. **2c**) decreased with depth while nutrient concentrations increased (Fig. **2d-e**). Nutrients were depleted (<0.5, <0.2, and <0.5 µmol L$^{-1}$ for DIN, PO$_4$, Si(OH)$_4$, respectively) in the mixed layer.

At the coastal stations (16.51 to 16.92 °W), the temperature had a range of 14.6-26.1 °C and salinity of 35.53-36.08 in the upper 150 m depth (Fig. **2a** & **b**). Here, the mixed layer was significantly shallower than in the open ocean (Tukey, $p<0.01$), with an average depth of $17 \pm 4$ m (SI Table **1**). Oxygen was decreasing with depth and a shallow oxygen minimum (OMZ; <50 µmol kg$^{-1}$) was detected (Fig. **2c**) from 80 m to 200 m depth. Nutrients (Fig. **2d-e**) were depleted at the surface (5 m depth) while the deeper coastal waters (~ 80 to 200 m depth) were colder and richer in nutrients than in the open ocean with on average 3.4 fold more nutrients (DIN, PO4, Si(OH)$_4$) when integrated over 100 m depth.

In the CE ('periphery' and 'core'), waters had a temperature range of 13.5-24.2 °C and salinity of 35.48-36.36 in the upper 150 m depth (Fig. **2a** & **b**). A tightening of isopycnals with a strong doming of the isotherms, isohalines, and nutriclines was observed (Fig. **2a-b**, **d-f**). A shallow OMZ was detected from ~30m to ~100 m depth with the lowest oxygen concentration (<10 µmol kg$^{-1}$) between 30-40 m depth. The mixed layer was significantly shallower (Tukey, $p<0.05$) at the CE periphery than in the open ocean, with an average of $15 \pm 6$ m depth. However, the CE core was not significantly different ($21 \pm 3$ m; Tukey, $p>0.05$). Nutrients **(**Fig. **2d-f)** were depleted (<0.5, <0.2 and <0.5 µmol L$^{-1}$ for DIN, PO$_4$, Si(OH)$_4$ respectively)  at the surface (~5 m) only in the Eastern (17.11 °W, 18 °N) and Western (18.83-19.11 °W, 18.58 °N) part of the CE periphery.

The Frontal Zone station E3 (19.55 °W) was distinct from the adjacent stations with respect to surface temperature (1 °C colder, Fig **2a**). A doming of the nutriclines was observed **(**Fig.**2d-f)** and nutrient concentrations integrated over 100 m depth at St. E3 were ~3 fold higher than Open ocean St. S4 (20.3 °W) and ~1.2 fold higher than CE periphery St. EDZ-1 (19.11 °W).



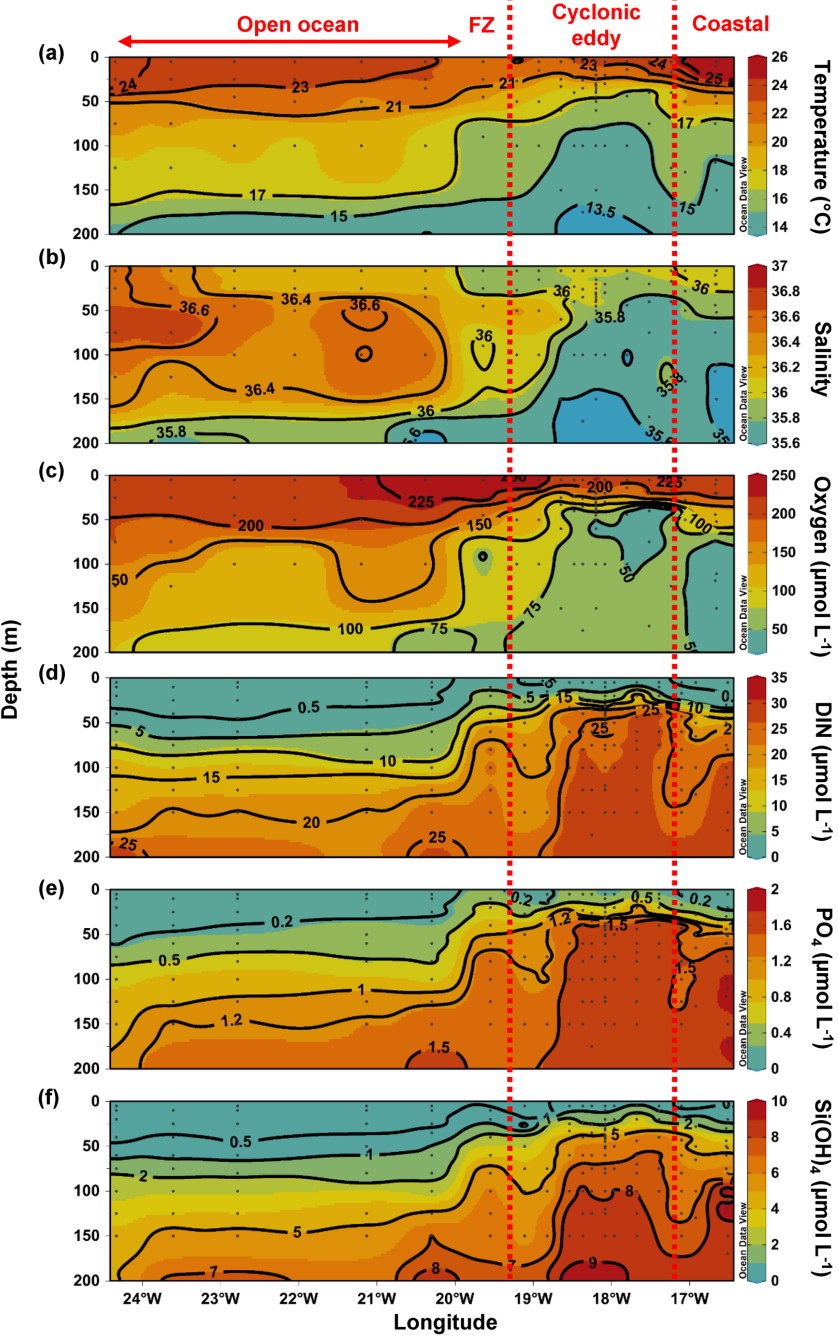

336

Figure 2: Epipelagic distribution (0-200m) of Temperature (**a**), Salinity (**b**), Oxygen (**c**), Total inorganic

nitrogen (DIN, **d**), $PO_4$ (**e**), $Si(OH)_4$ (**f**). Red dashed line show the cyclonic eddy periphery and FZ refer

as Frontal Zone.



### 3.2 Chlorophyll-*a* and primary production

In order to compare stations along the zonal transect and within the eddy, data were integrated over the water column (0-100 m depth). Along the zonal transect, depth-integrated Chl-*a* concentration ranged between 11.7 and 58.7 mg m$^{-2}$ and decreased from the coastal to the open ocean stations (Table **1**; SI Fig. **S4**). Depth-distribution (Fig. **3a**) presented a Chl-*a* maximum in the open ocean around ~75 m from 23.61 to 24.33 °W and around ~50 m from 22.78 to 20.3 °W, up to 0.70 µg L$^{-1}$. At the coastal stations, the Chl-*a* maximum was found between 30-40 m depth with values up to 0.96 µg L$^{-1}$. Integrated autotrophic plankton biomass (Table **1**) ranged between 1.6 and 7.8 and between 3.6 and 6.1 g C m$^{-2}$ in the open ocean and at the coastal stations, respectively. In the open ocean waters, autotrophic plankton biomass (Fig. **3b**) presented a gradient of distribution with a maximum around ~75 m from 23.61 to 24.33 °W, around ~50 m from 22 to 22.78 °W and between 5-25 m from 21.13 to 20.3 °W, with values up to 166 µg C L$^{-1}$. In the coastal stations, autotrophic plankton biomass maximum was found between 30-40 m depth with values up to 117 µg C L$^{-1}$. Both Chl-*a* concentration and autotrophic plankton biomass did not vary significantly between the open ocean and the coastal stations (Tukey, $p>0.05$). Integrated total and dissolved primary production (PP$_{TOT}$; PP$_{DOC}$; Table **1**) remained fairly constant with ranges of 101-137 and 42.8-78 mmol C m$^{-2}$ d$^{-1}$, respectively, from the coastal to the open ocean stations, except for the station furthest offshore (24.33 °W), where rates decreased sharply to 25.8 mmol C m$^{-2}$ d$^{-1}$ for PP$_{TOT}$ and to 12.3 mmol C m$^{-2}$ d$^{-1}$ for PP$_{DOC}$. The integrated percentage of extracellular release (PER; Table 1) in both regions ranged between 42.3-67.5%. Both PP$_{TOT}$ and PER did not vary significantly between the open ocean and the coastal stations (Tukey, $p>0.05$). PP$_{TOT}$ was decreasing with depth (Fig. **3c**) while PER was increasing (Fig. **3d**). In general, PP$_{TOT}$ and PP$_{DOC}$ were positively correlated to the Chl-*a* concentration (R$^2$=0.48 and 0.42 respectively; $p<0.001$; Fig. **6c & d**).

In the CE (core and periphery) and at the Frontal Zone integrated Chl-*a* concentration ranged from 17.2 to 225 mg m$^{-2}$ (Table **1**). The Chl-*a* distribution (SI Fig. **S4**) showed a clear spatial separation with the highest values (98.7-225 mg m$^{-2}$) in the western (18.83-19.11 °W, 18.29 °N) and northern (148 mg m$^{-2}$; 18.08 °W, 19.15 °N) part of the CE and lowest values (26.8-37.5 mg m$^{-2}$) in the eastern in the Southern (18.08 °W, 18 °N) and Eastern part (17.39 - 17.68 °W, 18.58 °N). Depth distribution of Chl-*a* concentration also differed across the eddy, with values >0.5 µg L$^{-1}$ reaching down to 45 m depth at the Frontal Zone and the western part of the CE (19.11-19.55 °W) and down to 30 m depth in the eastern side of the CE (17.1-17.4 °W). Within the upper 30 m, Chl-*a* concentration within the CE was significantly higher than at the



open ocean and the coastal stations (ANOVA, $p<0.05$). Integrated autotrophic plankton
biomass ranged between 0.3 and 4.7 g C m$^{-2}$ in the CE (Table **1**). Depth distribution of
autotrophic plankton biomass (Fig. **3b**) showed low biomass in the upper 40 m (<25 µg C L$^{-1}$)
from 18.83 to 19.11 °W. In contrast, higher biomass (>25 µg C L$^{-1}$) occurred in the more eastern
stations of the CE (17.11 to 18.54 °W) and westwards from the Frontal Zone (19.55 °W). In the
eddy, autotrophic plankton biomass reached higher concentrations mostly within the upper 40
m, with values up to 191 µg C L$^{-1}$. It should be noted that autotrophic biomass refers only to
pico- and nanophytoplankton and not to larger cells such as typical for diatoms or
dinoflagellates. Depth-integrated $PP_{TOT}$ and $PP_{DOC}$ rates were significantly higher in the CE and
at the Frontal Zone than at the open ocean and the coastal stations (Tukey, $p<0.05$) with values
ranging from 245 to 687 mmol C m$^{-2}$ d$^{-1}$ and from 95.9 to 238 mmol C m$^{-2}$ d$^{-1}$, respectively
(Table **1**). $PP_{TOT}$ rates (Fig. **2c**; Table **2**) were fairly constant across the CE's surface (5 m
depth), ranging between 11.7 to 13.3 µmol C L$^{-1}$ d$^{-1}$, but varied strongly between 15-40 m depth
with values from 0.2 to 14.5 µmol C L$^{-1}$ d$^{-1}$. The highest $PP_{TOT}$ rates were found in the Frontal
Zone with up to 25.0 µmol C L$^{-1}$ d$^{-1}$ at the surface. The range of $PP_{DOC}$ rates (Table **2**) was
larger in the CE (0.2-4.9 µmol C L$^{-1}$ d$^{-1}$) and the Frontal Zone (0.7-7.8 µmol C L$^{-1}$ d$^{-1}$) than in
the open ocean and at the coastal stations. Integrated PER had a range of 29.4-43.3 % (Table
**1**). A slightly lower PER was observed within the upper 40 m (Fig. **2d**) for the CE and Frontal
Zone compared to open ocean and coastal stations.

Table 1: Chlorophyll a (Chl *a*) and abundance, biomass and activity of phyto- and bacterial plankton,
integrated over the upper 100m depth. '-' indicate that the parameter was not measured. $PP_{DOC}$ and $PP_{TOT}$
rates in St EDM-4E were measured on the 22/07/2019 from 5, 33 and 50m depth and CR and BR rates
were measured in St. E5 on the 29/07/2019 from 5, 35 and 50m depth.

| Location | Station | Chl *a* (mg m$^{-2}$) | AutPl (g C m$^{-2}$) | $PP_{DOC}$ (mmol C m$^{-2}$ d$^{-1}$) | $PP_{TOT}$ (mmol C m$^{-2}$ d$^{-1}$) | PER (%) | HB ($10^{15}$ cell m$^{-2}$) | CR (mmol C m$^{-2}$ d$^{-1}$) | BR (mmol C m$^{-2}$ d$^{-1}$) | BP (mmol C m$^{-2}$ d$^{-1}$) |
|---|---|---|---|---|---|---|---|---|---|---|
| Coastal | E5 | 54.5 | 6.1 | 75.2 | 137 | 54.9 | 14.7 | 99.6 | 32 | 2.9 |
| | EDZ-10N | 36.8 | 3.6 | - | - | - | 13.8 | - | - | 4.1 |
| | AZM-3 | 58.7 | 5.3 | - | - | - | 12.9 | - | - | 5.7 |
| Eddy Periphery | EDZ-8N | 61.5 | 4.7 | - | - | - | 10.7 | - | - | 8.2 |
| | EDZ-7N | 26.8 | 1.6 | - | - | - | 9.4 | - | - | 5.7 |
| | EDZ-6N | 27.9 | 1.2 | - | - | - | 9.1 | - | - | 4.0 |
| Eddy Core | EDZ-5N | 39.2 | 4.1 | - | - | - | 14.5 | 154 | 59.1 | 4.7 |



Table 1 cont.: Chlorophyll a (Chl *a*) and abundance, biomass and activity of phyto- and bacterial
plankton, integrated over the upper 100m depth. '-' indicate that the parameter was not measured. $PP_{DOC}$
and $PP_{TOT}$ rates in St EDM-4E were measured on the 22/07/2019 from 5, 33 and 50m depth and CR and
BR rates were measured in St. E5 on the 29/07/2019 from 5, 35 and 50m depth.

| Location | Station | Chl *a* (mg m⁻²) | AutPl (g C m⁻²) | $PP_{DOC}$ (mmol C m⁻² d⁻¹) | $PP_{TOT}$ (mmol C m⁻² d⁻¹) | PER (%) | HB (10¹⁵ cell m⁻²) | CR (mmol C m⁻² d⁻¹) | BR (mmol C m⁻² d⁻¹) | BP (mmol C m⁻² d⁻¹) |
|---|---|---|---|---|---|---|---|---|---|---|
| Eddy Core | EDM-4E | 46.0 | 3.3 | 95.9 | 245 | 39.2 | 15.2 | 135 | 60.8 | 4.5 |
| | EDM-3E | 77.5 | 3.2 | - | - | - | 15.3 | - | - | 8.6 |
| | EDM-4 | 63.8 | 3.3 | 141 | 380 | 37.2 | 19.4 | 275 | 127 | 6.4 |
| Eddy Periphery | S5 | 35.7 | 3.6 | 117 | 288 | 40.8 | 23.7 | - | - | 6.8 |
| | EDM-5E | 35.2 | 1.6 | - | - | - | 11.8 | - | - | 4.7 |
| | EDM-2E | 148 | 1.7 | - | - | - | 20.8 | - | - | 11.4 |
| | EDZ-4 | 47.8 | 1.0 | - | - | - | 14.4 | - | - | 6.3 |
| | EDZ-3 | 17.2 | 0.3 | - | - | - | 9.6 | - | - | 2.9 |
| | EDZ-2 | 98.7 | 0.7 | 131 | 445 | 29.4 | 8.2 | 592 | 320 | 8.1 |
| | EDZ-1 | 225 | 0.6 | - | - | - | 13.7 | - | - | 19.3 |
| Frontal Zone | E3 | 72.1 | 2.4 | 238 | 687 | 34.6 | 12.9 | 529 | 257 | 7.7 |
| Open ocean | S4 | 40.2 | 4.5 | - | - | - | 16.9 | - | - | 4.3 |
| | S3 | 30.7 | 4.0 | 42.8 | 101 | 42.3 | 14.5 | 346 | 148 | 2.6 |
| | E2 | 22.3 | 4.4 | 78.0 | 116 | 67.5 | 12.2 | 387 | 168 | 2.3 |
| | S2 | 34.1 | 7.8 | - | - | - | 13.9 | - | - | 2.1 |
| | S1 | 12.2 | 1.6 | - | - | - | 5.4 | - | - | 0.7 |
| | E1 | 11.7 | 2.3 | 12.3 | 25.8 | 47.6 | 6.7 | 19.7 | 6.3 | 0.8 |


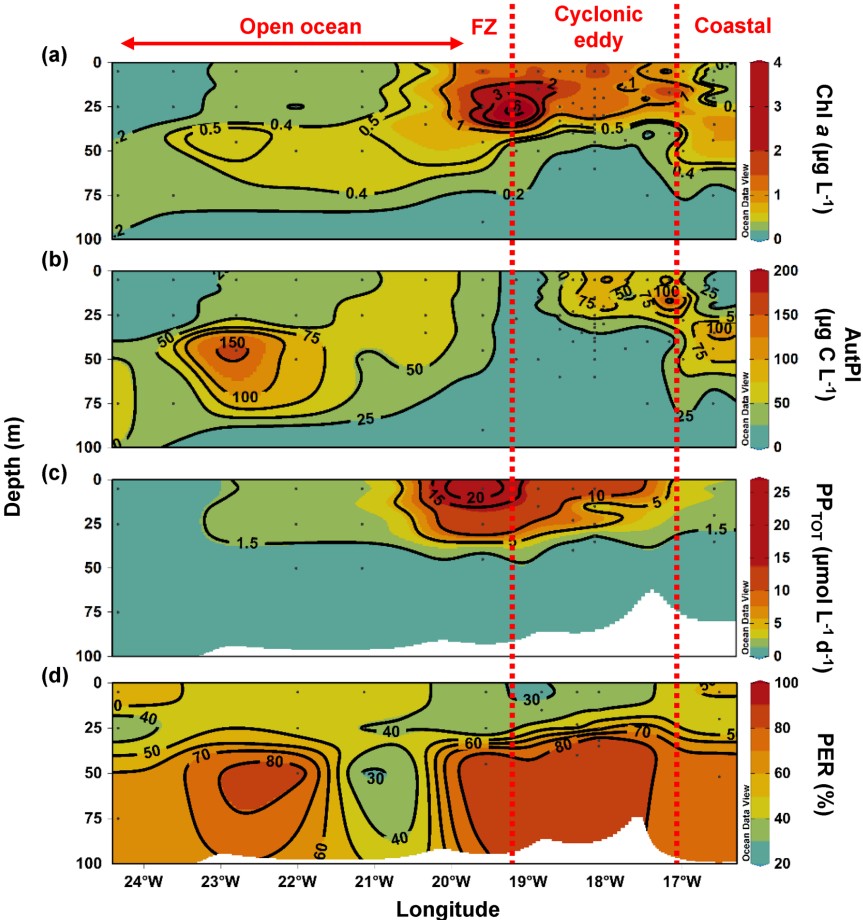


Figure 3: Depth distribution of phytoplankton biomass and activity over 100m depth: Chlorophyll *a* (Chl *a*; **a**), Autotrophic plankton biomass (AutPl; **b**), total primary production (PP$_{TOT}$; **c**), and percentage of extracellular release (PER; **d**). Red dashed line show the eddy-influenced area and FZ refer as Frontal Zone.

407

## 3.3 Bacterial abundance and activities

Heterotrophic bacterial abundance decreased with depth and was highest in the upper 50 m of all stations (Fig. **4a**). At the coastal and open ocean stations, integrated (0-100 m depth) heterotrophic bacteria abundance ranged between 12.9-14.7 and 5.4-16.9x10$^{15}$ cells m$^{-2}$, respectively (Table **1**). No significant differences in heterotrophic bacterial abundance were observed between the open ocean and coastal stations (Tukey, *p*>0.05). In the open ocean



waters, the lowest integrated BR and CR rates (Table 1) were reported at the station furthest
offshore (24.33 °W), with 6.3 and 19.7 mmol C m$^{-2}$ d$^{-1}$, respectively. Yet in the other open
ocean stations (21.13 to 22 °W), integrated BR and CR rates were higher (148-168 and 346-
348 mmol C m$^{-2}$ d$^{-1}$ respectively) than in the coastal station (32 and 98 mmol C m$^{-2}$ d$^{-1}$
respectively). Overall, BR and CR rates were higher in the open ocean than at the coastal
stations with high rates (> 1 and > 2.5 µmol C L$^{-1}$ d$^{-1}$, respectively) down to 60 m depth (Fig.
**4b**; SI Fig. **S5a**). Integrated BP, in contrast, was generally higher at the coastal stations with
2.9-5.7 mmol C m$^{-2}$ d$^{-1}$ compared to the open ocean with 0.7-4.3 mmol C m$^{-2}$ d$^{-1}$ (Table **1**).
However, BP rates were not significantly different from the open ocean (Tukey $p$>0.05), where
BP rates were more variable. At the coastal stations, the highest BP (Fig. **4b**) rates were
observed at the surface (5 m) and around ~40 m depth, while in the open ocean, the highest
rates were found at the surface (5 m). BGE was determined for the upper 50 m (Table **2**) and
showed only little variability over depth. However, BGE was significantly higher (Tukey, $p <$
0.05) at the coastal than at the open ocean stations with ranges of 5.3 ± 2.2 to 8.0 ± 1.0%
compared to 0.9 ± 0.04 to 2.3 ± 0.02%, respectively. We estimated the predominance of
autotrophy/heterotrophy in the system, by dividing the PP$_{TOT}$ rates by the BCD. Heterotrophic
conditions ($\frac{PP_{TOT}}{BCD}$ <1) occurred at the open ocean stations throughout the water column, while
autotrophic conditions ($\frac{PP_{TOT}}{BCD}$ >1) prevailed at the coastal St. E5 (Table **2**). This pattern was
preserved when data were integrated over the mixed layer (Fig. **5**) apart for the furthest station
offshore (24.33 °W) where autotrophy occurred, yet lower than at the coastal station St.E5
($\frac{PP_{TOT}}{BCD}$ = 2 and 5.5 respectively). PP$_{DOC}$ rates were sufficient to satisfy the BCD at the coastal
St.E5 but not in the open ocean stations (Table **2**).
In the CE and at the Frontal Zone, integrated heterotrophic bacterial abundance ranged from
8.2 - 23.7x10$^{15}$ cells m$^{-2}$ (Table **1**). In the CE, substantial variation of bacterial abundance
occurred within the upper 20 m (Fig. **4a**), with an abundance of <1x10$^9$ cells L$^{-1}$ in the western
CE periphery (18.83 to 19.11 °W) and > 3x10$^9$ cells L$^{-1}$ in the CE core stations (~18 °W).
Depth-integrated BR and CR (Table **1**) ranged between 59.1 and 320 and between 135 and 592
mmol C m$^{-2}$ d$^{-1}$, respectively. Elevated BR and CR rates (> 1 and 2.5 µmol C L$^{-1}$ d$^{-1}$,
respectively) were only present in the upper ~30-40 m of the CE (Fig. **4b**; SI Fig. **S5a**).
Integrated BP rates ranged from 2.9 to 19.3 mmol C m$^{-2}$ d$^{-1}$ in the CE and at the Frontal Zone
stations (Table **1**). BP rates in the upper 40 m of the CE and at the Frontal Zone were elevated
but were significantly higher than in the coastal and open ocean stations only in the stations



within the CE periphery (Tukey $p<0.05$). Stations in the core of the CE had BGEs (Table **2**)
significantly higher than the stations located in the open ocean (Tukey, $p<0.05$). BGE had a
range of $1.4 \pm 2.2$ to $10.5 \pm 0.5$ % and $2.8 \pm 0.1$ to $3.0 \pm 1.7$% in the CE and the Frontal Zone
stations, respectively. Highest BGE was observed below 20 m depth in the CE core (up to
10.48%, St EDM-4E). With ratios ranging from 1.13 to 3.5, the upper 40 m of the CE and the
Frontal Zone stations were rather autotrophic (Table **2**). When integrated over the mixed layer
(Fig. **5**), stations within the CE and at the Frontal Zone were autotrophic, with a $\frac{PP_{TOT}}{BCD}$ ratio
ranging from 1.17 to 3.8. $PP_{DOC}$ was on average 70% of the BCD within the CE and the Frontal
Zone, yet ranging from 28.3 to 114.5%.

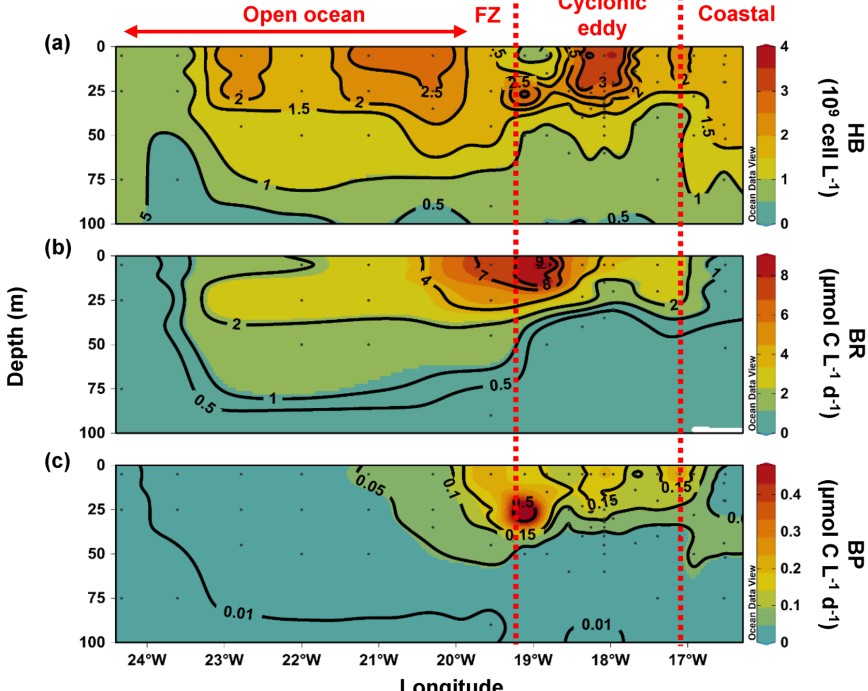


Figure 4: Depth distribution of bacterial abundance and microbial activities over 100m depth:
Heterotrophic bacterial abundance (HB; **a**), bacterial respiration (BR; **b**), bacterial production (BP; **c**).
Red dashed line show the eddy-influenced area and FZ refers to Frontal Zone.




Table 2: Average (mean) ± standard deviation of microbial metabolic activities during M156: bacterial
carbon demand (BCD); bacterial growth efficiency (BGE); dissolved primary production (PP$_{DOC}$);
Percentage of extracellular release (PER); total primary production (PP$_{TOT}$) and the ratio between BCD
and PPTOT ($\frac{BCD}{PP_{TOT}}$). BCD and BGE were obtained from BP and BR rates at 22°C (see text). '-' indicate
that the parameter was not measured and B.D. below detection (see text). PP$_{DOC}$ and PP$_{TOT}$ rates in St.
EDM-4E were measured on the 22/07/2019 from 5, 33 and 50m depth and CR and BR rates were
measured in St. E5 on the 29/07/2019 from 5, 35 and 50m depth.

| Location | Station | Depth (m) | BCD (µmol C L$^{-1}$ d$^{-1}$) | BGE (%) | PP$_{DOC}$ (µmol C L$^{-1}$ d$^{-1}$) | PER (%) | PP$_{TOT}$ (µmol C L$^{-1}$ d$^{-1}$) | $\frac{BCD}{PP_{TOT}}$ |
|---|---|---|---|---|---|---|---|---|
| Coastal | E5 | 5 | 0.6 ± 0.1 | 5.3 ± 2.2 | 1.5 ± 0.2 | 34.9 ± 1.1 | 2.7 ± 0.2 | 4.5 ± 1.5 |
| | | 20 | 0.5 ± 0.1 | 6.9 ± 1.6 | 1.2 ± 0.1 | 52.6 ± 2.7 | 2.5 ± 0.1 | 5.5 ± 1.4 |
| | | 35 | 0.5 ± 0.3 | 8.0 ± 1.0 | 0.7 ± 0.1 | 89.8 ± 3.9 | 1.0 ± 0.1 | 2.1 ± 0.2 |
| | EDZ-10N | All | - | - | - | - | - | - |
| | S6 | All | - | - | - | - | - | - |
| Eddy Periphery | EDZ-8N | All | - | - | - | - | - | - |
| | EDZ-7N | 5 | 3.5 ± 0.7 | 3.6 ± 0.3 | - | - | - | - |
| | | 20 | 3.5 ± 0.3 | 3.3 ± 1.7 | - | - | - | - |
| | EDZ-6N | All | - | - | - | - | - | - |
| Eddy Core | EDZ-5N | 5 | 2.6 ± 0.4 | 6.02 ± 1.5 | - | - | - | - |
| | | 20 | 1.15 ± 0.3 | 9.51 ± 2.1 | - | - | - | - |
| | | 30 | 0.41 ± 0.6 | 7.11 ± 0.2 | - | - | - | - |
| | | 100 | B.D. | B.D. | - | - | - | - |
| | EDM-4E | 5 | 4.5 ± 0.4 | 4.1 ± 1.1 | 4.3 ± 0.1 | 36.7 ± 0.2 | 11.2 ± 0.1 | 2.5 ± 0.2 |
| | | 15 | 1.3 ± 0.4 | 10.5 ± 0.6 | 0.4 ± 0.1 | 39.3 ± 6.8 | 1.1 ± 0.1 | 2.1 ± 0.4 |
| | | 35 | B.D. | B.D. | 0.6 ± 0.3 | 94.4 ± 0.9 | 0.6 ± 0.3 | - |
| | | 60 | B.D. | B.D. | - | - | - | - |
| | EDM-3E | All | - | - | - | - | - | - |
| | EDM-4 | 5 | 4.7 ± 1.1 | 3.2 ± 1.4 | 4.3 ± 1.0 | 35.1 ± 5.7 | 12.6 ± 1.2 | 2.7 ± 1.1 |
| | | 23 | 3.4 ± 0.2 | 4.4 ± 2.1 | 3.9 ± 0.2 | 35.7 ± 1.4 | 11.0 ± 0.3 | 3.2 ± 1.4 |
| | | 40 | B.D. | B.D. | 0.3 ± 0.1 | 85.3 ± 7.1 | 0.3 ± 0.1 | - |
| | | 100 | B.D. | B.D. | - | - | - | - |
| Eddy Periphery | S5 | 5 | - | - | 4.8 ± 0.4 | 34.9 ± 1.1 | 13.7 ± 0.7 | - |
| | | 25 | - | - | 3.4 ± 0.3 | 52.6 ± 2.7 | 6.5 ± 0.4 | - |
| | | 32 | - | - | 0.2 ± 0.1 | 89.8 ± 3.9 | 0.2 ± 0.1 | - |





Table 2 cont.: Average (mean) ± standard deviation of microbial metabolic activities during M156:
bacterial carbon demand (BCD); bacterial growth efficiency (BGE); dissolved primary production
(PP$_{DOC}$); Percentage of extracellular release (PER); total primary production (PP$_{TOT}$) and the ratio
between BCD and PPTOT ($\frac{BCD}{PP_{TOT}}$). BCD and BGE were obtained from BP and BR rates at 22°C (see
text). '-' indicate that the parameter was not measured and B.D. below detection (see text).

| Location | Station | Depth (m) | BCD (µmol C L$^{-1}$ d$^{-1}$) | BGE (%) | PP$_{DOC}$ (µmol C L$^{-1}$ d$^{-1}$) | PER (%) | PP$_{TOT}$ (µmol C L$^{-1}$ d$^{-1}$) | $\frac{BCD}{PP_{TOT}}$ |
|---|---|---|---|---|---|---|---|---|
| Eddy Periphery | EDM-5E | All | - | - | - | - | - | - |
| | EDM-2E | All | - | - | - | - | - | - |
| | EDZ-4 | All | - | - | - | - | - | - |
| | EDZ-3 | All | - | - | - | - | - | - |
| | EDZ-2 | 5 | 10.5 ± 0.5 | 1.4 ± 2.2 | 2.9 ± 0.3 | 25.1 ± 3.4 | 11.9 ± 1.0 | 2.1 |
| | | 15 | 9.4 ± 2.3 | 2.5 ± 0.7 | 4.9 ± 0.1 | 31.0 ± 1.7 | 14.5 ± 0.6 | 0.3 |
| | | 50 | B.D. | B.D. | - | - | - | - |
| | | 100 | B.D. | B.D. | - | - | - | - |
| | EDZ-1 | All | - | - | - | - | - | - |
| Frontal Zone | E3 | 5 | 7.1 ± 0.4 | 3.0 ± 1.7 | 7.8 ± 0.4 | 31.7 ± 1.7 | 25.0 ± 0.9 | 3.5 ± 2.2 |
| | | 25 | 4.8 ± 1.1 | 2.8 ± 0.1 | 5.0 ± 0.6 | 33.4 ± 3.2 | 14.3 ± 0.8 | 3.0 ± 0.7 |
| | | 45 | 1.9 ± 0.6 | 2.9 ± 2.1 | 0.7 ± 0.2 | 87.0 ± 3.3 | 0.8 ± 0.2 | 0.4 ± 0.3 |
| | | 90 | B.D. | B.D. | - | - | - | - |
| Open ocean | S4 | All | - | - | - | - | - | - |
| | S3 | 5 | 3.2 ± 0.5 | 1.6 ± 0.2 | 1.3 ± 0.2 | 49.1 ± 5.5 | 2.7 ± 0.3 | 0.9 ± 0.5 |
| | | 25 | 2.6 ± 0.5 | 1.7 ± 1.1 | 1.16 ± 0.03 | 38.4 ± 0.9 | 2.5 ± 0.03 | 1.0 ± 0.3 |
| | | 50 | 1.2 ± 1.1 | 1.8 ± 0.2 | 0.0 ± 0.01 | 21.8 ± 6.6 | 0.1 ± 0.01 | 0.1 ± 0.1 |
| | | 100 | B.D. | B.D. | - | - | - | - |
| | E2 | 5 | 1.8 ± 0.6 | 1.8 ± 0.2 | 0.6 ± 0.1 | 40.9 ± 3.4 | 1.38 ± 0.1 | 0.8 ± 0.1 |
| | | 25 | 3.5 ± 1.1 | 0.9 ± 0.04 | 0.94 ± 0.1 | 50.2 ± 3.1 | 1.89 ± 0.1 | 0.5 ± 0.1 |
| | | 50 | 1.7 ± 0.4 | 1.6 ± 0.4 | 1.25 ± 0.3 | 91.3 ± 2.5 | 1.4 ± 0.3 | 0.8 ± 0.8 |
| | | 100 | B.D. | B.D. | - | - | - | - |
| | S2 | All | - | - | - | - | - | - |
| | S1 | All | - | - | - | - | - | - |
| | E1 | 5 | 0.4 ± 0.2 | 2.3 ± 0.02 | 0.23 ± 0.1 | 54.7 ± 13.3 | 0.39 ± 0.1 | 0.9 ± 0.5 |
| | | 25 | B.D. | B.D. | 0.18 ± 0.01 | 38.5 ± 0.6 | 0.43 ± 0.01 | - |
| | | 75 | B.D. | B.D. | 0.08 ± 0.02 | 61.7 ± 6.2 | 0.13 ± 0.02 | - |
| | | 125 | B.D. | B.D. | - | - | - | - |







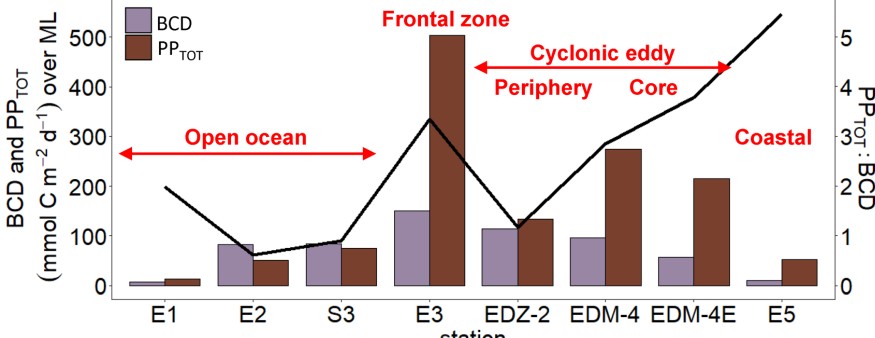


Figure 5: Integrated total primary production (PP$_{TOT}$) and bacterial carbon demand (BCD) rates over the
mixed layer during M156. Blackline reports the ratio between PP$_{TOT}$ and BCD. More information are
given in SI table 1.

3.4 Indices of phyto- and bacterioplankton activity change
We investigated the impact of the CE on heterotrophic bacterial and phytoplankton abundance
by regression analysis of, cell-specific BR and BGE (Fig. **6a**), as well as autotrophic plankton
biomass and Chl-*a* (Fig. **6b**). We noticed a negative semilogarithmic relationship (Fig. **6a**)
between cell-specific BR rates and the BGE in both the zonal transect (coastal+open ocean)
[BG= -3.11 ln (cell-specific BR) + 2.35; R²=0.86; *p*<0.001] and the eddy influenced region (CE
+ Frontal Zone) [BGE= -1.92 ln (cell-specific BR) + 5.28; R²=0.70; *p*=0.001]. Concerning the
phytoplankton (Fig. **6b**), we observed that Chl-*a* and autotrophic plankton biomass were
linearly correlated in the open ocean and coastal region (R²=0.75; *p*<0.001) while being poorly
correlated in the CE-influenced area (R²=0.13).



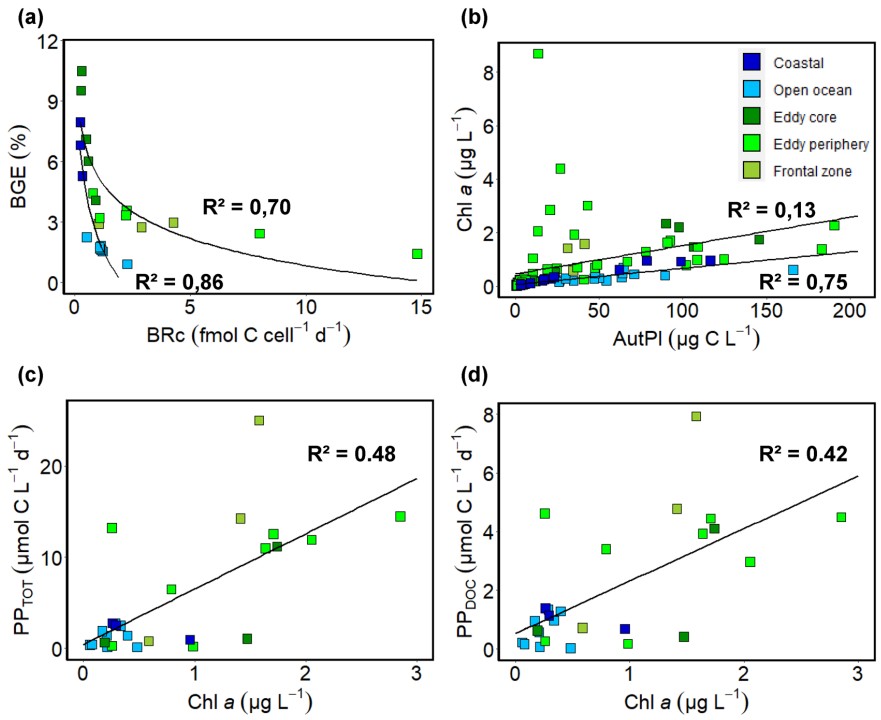


Figure 6: Relationship between (**a**) cell-specific bacterial respiration (BRc) and bacterial growth
efficiency (BGE), (**b**) chlorophyll *a* (Chl *a*) and autotrophic plankton biomass (AutPl), (**c**) total primary
production ($PP_{TOT}$) and Chl *a* and (**d**) dissolved primary production ($PP_{DOC}$) and Chl *a*. Black lines in
(**a**) and (**b**) show regression from the open ocean and coastal stations (blue shades) and from the stations
in eddy influenced area (green shades). Black lines in (**c**) and (**d**) show regressions in all the stations.


## 3.5 Semi-labile dissolved organic carbon

Between coastal and open ocean stations, SL-DOC concentration was not significantly different
(Tukey, *p*>0.05; SI Fig. **S5b**) with ranges of 1.9-8.0 µmol L$^{-1}$ and 4.7-18.9 µmol L$^{-1}$,
respectively. At those sites, SL-DOC distribution was rather uniform in the upper 40 m with
SL-DOC > 5 µmol L$^{-1}$, apart from the station furthest offshore from 22.7-24.3 °W where SL-
DOC > 5 µmol L$^{-1}$ was limited to shallow depth (5 m). In the CE and at the Frontal Zone, SL-
DOC concentration was clearly elevated and increased from East to West with an overall range
of 1.4-54.3 µmol L$^{-1}$. At the Frontal Zone, SL-DOC concentration > 5 µmol L$^{-1}$ was detectable
down to 90 m depth.



### 3.6 Correlation analysis

We applied a Pearson correlation matrix (Fig. **7**) to reveal significant correlations between the measured parameters. Temperature correlated negatively with nutrients (DIN, $PO_4$, $Si(OH)_4$; Pearson, $R < -0.9$, $p < 0.001$) and positively with bacteria (Pearson, $R = 0.65$, $p < 0.001$). Total ($PP_{TOT}$) and dissolved primary production ($PP_{DOC}$) were positively correlated to each other (Pearson, $R = 0.98$, $p < 0.001$) and to Chl-*a* and SL-DOC (Pearson, $R > 0.65$ and $> 0.60$ respectively, $p < 0.001$), but not to the autotrophic plankton biomass (Pearson, $R < 0.14$, $p > 0.05$). Bacterial biomass production (BP) and respiration (BR) were positively correlated (Pearson, $R = 0.78$, $p < 0.001$). BCD was more correlated to BR than to BP (Pearson, $R = 1$ and $R = 0.74$ respectively, $p < 0.001$). A clear coupling between phytoplankton and bacteria was indicated, by positive correlations between $PP_{TOT}$ and $PP_{DOC}$ and BP, BR, and BCD (Pearson, $R > 0.70$, $p < 0.001$), BP and Chl-*a* (Pearson, $R = 0.93$, $p < 0.001$), and BR and Chl-*a* and the SL-DOC concentration (Pearson, $R = 0.78$ and $0.75$ respectively, $p < 0.001$).



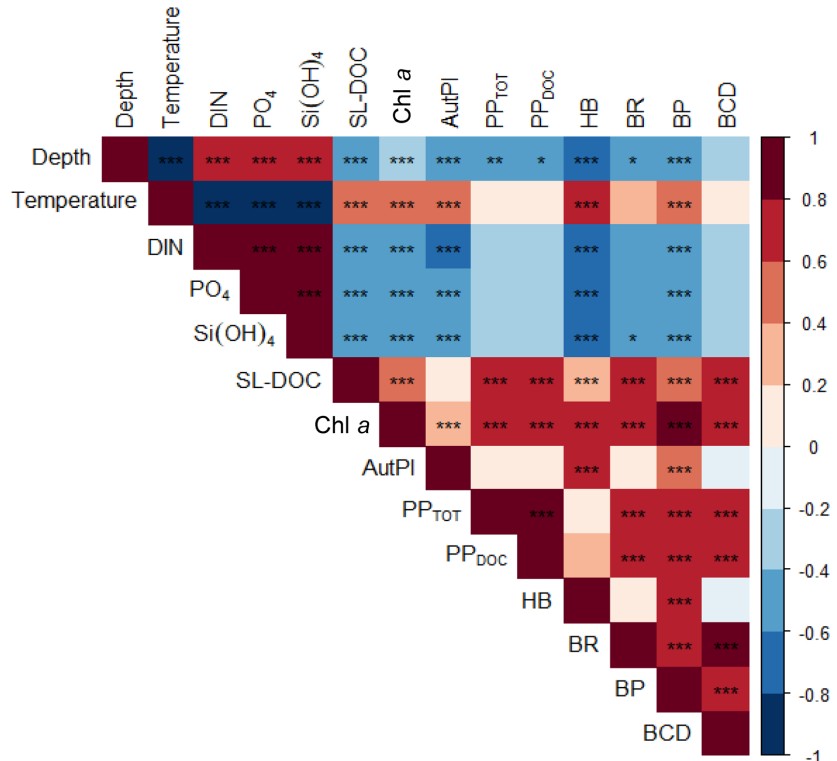

Figure 7: Correlations of biochemical parameters, metabolic activities, and bacterial abundance in the upper 200 m during M156. Colour scale: correlation coefficient (r). Statistical significance: '***'< 0.001, '**'< 0.01, '*'< 0.05.

## 4. Discussion

4.1 Distribution of phytoplankton abundance and activity in the Mauritanian upwelling system associated with cyclonic eddy perturbation

In general, coastal Chl-*a* concentration during this study was not as high as observed in earlier studies with strong coastal upwelling (e.g. Alonso-Sáez et al., 2007; Agustí and Duarte, 2013; Arístegui et al., 2020). This might be related to the relatively weak upwelling, as a result of weak surface winds along the Mauritanian Coast typically occurring during summer when our samples were collected (Peligrí and Peña-Izquierdo, 2015a). Consequently, during summer,



fewer nutrients reach the euphotic zone by coastal upwelling, while offshore surface wind remains strong and might enhance vertical mixing at the surface. Coastal Chl-*a* concentration was only slightly higher compared to the open ocean, and both the coastal and open ocean phytoplankton communities were dominated by cells <20µm, as indicated by the strong linear correlation between Chl-a and autotrophic plankton biomass (Fig. 6b).

We did not observe a marked gradient in phytoplankton productivity either, unlike other regions of the CanUS with permanent upwelling conditions (Demarcq and Somoue, 2015; Arístegui et al., 2020). $PP_{TOT}$ rates stayed rather constant from the coast to the open ocean and were in the range of reported rates in oligotrophic offshore waters of the CanUS (Agustí and Duarte, 2013; Lasternas et al., 2014). SL-DOC was relatively constant as well, with variations attributable to the westward propagation of the currents and eddies (SI Fig. **S5b**; Lovecchio et al., 2017, 2018). The absence of upwelling and the dominance of small autotrophic cells (<20µm) in the phytoplankton community suggest that in the open ocean and coastal stations, primary productivity was maintained through remineralisation of nutrients released from dying cells. Indeed, plankton mortality rates have been reported to increase with decreasing cell size (Marbá et al., 2007) and with increasing PER (Lasternas et al., 2014). Agustí and Duarte (2013) reported PER to range from ~1% in 'healthy' communities from the upwelled waters of the CanUS to ~70% in 'dying' communities from the oligotrophic waters of the ETNA. PER in our study was on average 51.1 ± 17% in the open ocean and coastal stations leading to the conclusion that primary productivity in those areas was maintained mainly through remineralisation of small (<20µm) plankton cells.

The CE broke this rather uniform distribution of phytoplankton productivity and community through coastal and open ocean waters. From a depth distribution perspective, Chl-*a* isolines seemed to have been pushed toward the surface in the CE (Fig. **3a**). Similar 'compression' of Chl-*a* isolines towards the surface have been reported in eddies earlier (Lochte and Pfannkuche 1987; Feng et al., 2007; Noyon et al., 2019). Such compressions have been attributed to resulting from phytoplankton growth through upwelling of nutrients combined with high vertical mixing from strong surface winds, which favour phytoplankton distribution at the surface (Feng et al., 2007; Noyon et al., 2019). In the CE, the upwelling was marked by the hydrographic parameters (e.g. temperature, salinity, nutrients, Fig. **2**), and before the eddy survey, strong surface winds occurred offshore (SI Fig. **S7**). Therefore, the phytoplankton which grew from upwelled nutrients must have been relocated to the surface through mixing,





the reason why high Chl-*a* (>0.5 μg L$^{-1}$) concentration was found at the surface (5m) in all
stations within the CE.
In addition, Chl-*a* was dispatched differently within the CE with the highest concentrations in
the Western and Northern part and lowest concentrations in the Southern and Eastern part
(Table **1**; SI Fig. **S4**). Furthermore, an almost continuous deepening of high Chl-a (>0.5 μg L$^{-1}$)
distribution, as well as an increase of SL-DOC concentration, was observed in the CE from
East to West (Fig. **3a**; SI Fig. **S5b**). Chelton et al. (2011) established from satellite observation
and an eddy-centric perspective that due to the rotational flow and the westward propagation of
CEs Chl-*a* tends to accumulate in their Southwest quadrants while being lower in their
Northeast quadrants. Since in our case, the CE shape was elliptic, we assume that the rotational
flow in the CE changed, shifting the accumulation. To the best of our knowledge, this is the
first time that high-resolution sampling could demonstrate this specific submesoscale Chl-*a*
distribution within a CE.
Outside of the CE boundaries, we noticed a thermal front with colder surface water. Thermal
fronts are often detected out of eddies periphery as a consequence of eddy-eddy interaction (See
review by Mahadevan, 2016) and/or eddy-wind interaction (Xu et al., 2019). In this Frontal
Zone, we observed higher nutrient content than the adjacent stations and a doming of the
nutriclines marking an upwelling (Fig. **2a**, **d-f**). Thus, Chl-*a* was elevated, and 'compressed' to
the surface similarly as in the CE (Fig. **3a**). We assume this distribution to be the consequence
of the same factors affecting the CE (upwelling, mixing induced by strong surface winds).
In the CE-influenced area (CE+Frontal Zone), Chl-*a* concentration was disconnected from
small (<20μm) autotrophic plankton biomass (Fig. **6b**). This implies that in the West of the
eddy where Chl-*a* was high and small autotrophic plankton biomass low (Fig. **3a & b**), larger
autotrophic cells such as diatoms and/or dinoflagellate were present in higher quantities. We
corroborate this point from lipid biomarkers concentration (unpublished data) as fucoxanthin,
a typical marker of diatoms (Stauber and Jeffrey, 1998), was the dominant pigment in the
Western part of the CE. This is consistent with previous studies in which CEs unevenly altered
the phytoplankton community, often reporting the presence of diatoms/dinoflagellates (e.g.,
Lochte and Pfannkuche, 1987; Lasternas et al., 2013). The details of autotrophic plankton
composition (SI Fig. **S7**) confirm this diversity, with the uneven distribution of cyanobacteria
(Synechococcus) and eukaryotic pico- and nanoplankton within the CE underscoring the fact
that the phytoplankton community was likely separate from the transect and diverse within a
submesoscale range.



Therefore, the CE dispatched different phytoplankton taxa with different potentials of primary
production and resources acquisition. Moreover, the mixed layer was also highly variable
within the CE leading to substantial variation of $PP_{TOT}$ rates (SI Table **1**, Figure **5**). Hence, we
observed a three-fold variation of depth-integrated $PP_{TOT}$ rates over 100m depth (Table **1**)
within the CE which is coherent with earlier observations of a fivefold variation of primary
production integrated over the euphotic zone in a CE in the subtropical Pacific Ocean
(Falkowski et al., 1991). Overall, primary productivity was enhanced within the CE and the
Frontal Zone with an average of fourfold more depth-integrated $PP_{TOT}$ rates over 100m depth
than in the open ocean and coastal stations. This is coherent with Löscher et al. (2015) who
found that depth-integrated primary productivity over the chlorophyll *a* maximum of a CE in
the Mauritanian upwelling system was threefold higher than the surrounding waters. Exudation
rates ($PP_{DOC}$) were also enhanced within the eddy and integrated (0-100 m) $PP_{DOC}$ rates were
on average three-fold time higher than in the transect (Table **1**). Yet, even if $PP_{DOC}$ rates were
higher within the CE and at the Frontal Zone stations (Table **2**), PER was slightly lower at the
surface (Fig. **3d**). We start from two hypotheses regarding this distribution 1) the lower PER
reported was due to a higher proportion of larger phytoplankton (e.g. diatoms) who have lower
turnover rates and therefore have lower PER and/or 2) the upwelling of nutrients generated by
the CE might have enhanced the physiological health of the phytoplankton community (Agustí
and Duarte, 2013; Laternas and Agustí, 2014).

4.2 Heterotrophic bacteria abundance and activities responses in the Mauritanian
upwelling system

Along the zonal transect (open ocean+coastal stations), a strong coupling between HB
abundance and $PP_{TOT}$ rates was observed ($R^2$=0.72). Therefore, HB abundance followed the
same trends as the $PP_{TOT}$ by being continuously distributed from the coast to the offshore
waters. Bachmann et al. (2018) reported a similar trend in the Mauritanian upwelling system
during summer, strengthening our finding.
Bacterial activities were distributed differently. Both BR and BP were within the range of
reported rates for coastal and offshore water of the CanUS (Reinthaler et al., 2006; Alonso-
Saez et al., 2007; Vaqué et al., 2014).BP rates slightly decreased from the coast to the open
ocean. Similar trends were found in the CanUS with different upwelling intensities and at



different seasons (Alonso-Saez et al., 2007; Vaqué et al., 2014). Therefore, those factors (upwelling intensity and seasonality) were likely only indirectly coupled with BP variability, which instead was rather driven by the composition of the phytoplankton community. Indeed, BP was more correlated to Chl-*a* than autotrophic plankton biomass (<20μm; Fig. **7**) suggesting that BP was more enhanced by the presence of larger autotrophic cells, such as diatoms or dinoflagellates. Those have larger phycospheres allowing them to attract more bacteria by chemotaxis (see review by Seymour et al., 2017). Hence, bacteria may benefit from mutualistic relationships with larger algae increasing their BP. Fucoxanthin, was decreasing from the coastal to offshore waters with overall low relative abundance (5-15%) (data not shown). Being part of microphytoplankton, especially diatoms have higher viability in coastal than in offshore waters of the CanUS (Lasternas et al., 2013), which may explain the observed fucoxanthin gradient.

In contrast, BR rates were higher in offshore than in coastal waters. BR rates were coupled to SL-DOC concentration, which is in agreement with Xu et al. (2013), who also found BR to be enhanced by low molecular weight DOC compound (<30kDa). SL-DOC compounds have a turnover of weeks to months, which allows them to escape rapid microbial degradation (Hansell et al., 2009). In the CanUS, currents and eddies can laterally transport DOC up to 2000 km (Lovecchio et al., 2018). Hence, we state that SL-DOC compounds produced at the coast have been relocated offshore while being slowly respired by heterotrophic bacteria along the way.

The distinct distribution of BP and BR rates affected the distribution of the BGE, which was higher in the coastal than in the open ocean stations. This is in accordance with observations by Alonso-Sáez et al. (2007) who showed higher BGE in the upwelling area above Cape Blanc than in the offshore waters of the CanUS. Overall, the BGEs reported here are among the lowest reported with all values <11%, but not surprising since BGE is negatively correlated to temperature and, therefore, reduced in the tropical ocean (Rivkin and Legendre, 2001). Yet we report an average BGE three times lower than Alonso-Sáez et al., (2007). We assume this difference to result from the difference in upwelling intensity (none vs. permanent). Indeed, Kim et al. (2017) denoted that BGE increased with increasing upwelling intensity in the Ulleung Basin. Under none or low upwelling conditions, bacteria compete with phytoplankton for nutrient acquisition. Moreover, as microphytoplankton do not thrive in the water column due to their high nutrient requirements (see review by Marañón, 2015), bacteria benefit less from their phycospheres. Hence, we expect BP to be lower in the relaxation period (May to July)



post upwelling than in the upwelling season (January to March; Lathuilière et al., 2008) in the
Mauritanian upwelling system.
Within the CE-influenced stations (CE + Frontal Zone), HB abundance was disconnected from
the $PP_{TOT}$ rates (Fig. **4a**). HB abundance was significantly higher in the core of eddy but
surprisingly low at the Southwestern side of the eddy periphery (18.83 to 19.11 °W), where
both $PP_{TOT}$ rates and Chl-*a* were high (Fig. **3a**, **c**). Hernández-Hernández et al. (2020) reported
a similar feature with a strong disparity of HB biomass distribution within a CE in the CanUS.
Since Chl-*a* and SL-DOC compounds accumulated in the Southwestern part of the CE, gel-
likes particles produced by phytoplankton and bacteria such as transparent exopolymer particles
(TEP) (Passow, 2002) might have also accumulated there. We hypothesize that a missing
fraction of the bacteria might have been attached to gel-like particles (Busch et al., 2018) or
other particulate matter.
The BP was particularly stimulated within the CE-influenced stations and on average threefold
higher than in the open ocean stations when integrated over 100 m. This is in accordance with
earlier studies from the Sargasso Sea (Ewart et al., 2008), the CanUS (Baltar et al., 2010), and
in the Mediterranean Sea (Belkin et al., 2022) where CEs enhanced BP. As stated previously,
the upwelling induced by the CE and the Frontal Zone led to higher phytoplankton biomass,
including diatoms and/or dinoflagellates which were likely responsible for this increase in BP.
BR rates were also enhanced at the surface of the CE and were coupled to the SL-DOC
concentration. Since the CE was relatively young (1.5 months old), autochthonous SL-DOC
compounds produced by exudation ($PP_{DOC}$) must have been merged with allochthonous coastal
SL-DOC compounds transported during the CE formation. $PP_{DOC}$ rates in the CE covered 28.3
to 114.5% of the BCD, indicating a moderate to strong trophic dependence of bacteria on
phytoplankton in CE (Fouilland and Mostajir, 2010). Although $PP_{TOT}$ may satisfy the BCD in
the CE through the bacterial incorporation of phytoplankton-derived DOC from sloppy feeding,
exudation, viral infection, or cell apoptosis, a question remains about why heterotrophs
preferentially used SL-DOC compounds for respiration rather than for biomass production. We
start from two hypotheses, firstly, the SL-DOM compounds had a high C/N ratio leading to an
increase of BR and a decrease of BGE (Lønborg et al., 2011). Secondly, SL-DOC was easier to
access for bacteria than other nutrients. Phytoplankton-DOM exudate/lysates are more or less
labile following their origin (e.g. diatoms/cyanobacteria) and are depleted in the nutrient (e.g.
nitrate/phosphate) limiting phytoplankton growth (e.g. Pete et al., 2010; Wear et al., 2020). As
the phytoplankton community was diverse within the CE and as the CE likely transported



allochthonous DOM, a multitude of compounds with specific qualities coexisted in the CE.
Therefore, bacteria may have used SL-DOC as fuel to degrade DOM compounds containing
limiting nutrients for their growth (Guillemette et al., 2016).
The diversity of DOM from different origins (e.g. cyanobacteria/diatom) within the CE likely
induced distinct bacterial communities. We noticed a negative semilogarithmic relationship
(Fig **6**) between cell-specific BR and the BGE in both the zonal transect (coastal+open ocean
stations) and the CE influenced (CE + Frontal Zone) stations. The slopes of the curves and the
ranges of cell-specific BR values were different between the two systems suggesting distinct
bacterial communities with different degrees of resource optimization (Baña et al., 2014).
Within the CE, the bacterial community was probably as the phytoplankton community even
more diverse as observed in previous CEs studies (Zhang et al., 2011; Yan et al., 2018).
Our results show that bacteria do not grow proportionally to the amount of DOM they received
through exudation but rather depends on the different requirement between respiration and
biomass production. In response, the BGE varied sevenfold within the CE (1.4-10.5%) whereas
it varied twofold in the open ocean (0.9-2.3%) and in the coastal (5.3-7.9%) stations. Robinson
(2008) suggested that most of the BGE variability within oligotrophic waters is explained by
BR. Here we hypothesise that in CEs, which cross oligotrophic waters in the ETNA, BGE
variability depends on both BP through phytoplankton taxonomical composition and BR
through the amount and quality of the SL-DOC.
Overall, we showed that autotrophy prevails in the upper 100m depth of Mauritanian coastal
waters while heterotrophy prevailed offshore. This is coherent with a modeling study from
Lovecchio et al. (2017). The CE and the associated Frontal Zone fuelled phytoplankton
nutrients needs and maintained autotrophy offshore. The highest $PP_{TOT}$ and the most
pronounced autotrophy were determined at the Frontal Zone. Mouriño-Carballido (2009)
reported from indirect estimations of net community production that the frontal zones between
CEs and ACEs are among the most productive area in the North West subtropical Atlantic
Ocean. Previous studies showed that the trophic balance could switch from autotrophy to
heterotrophy in an eddy within a month(s) (Maixandeau et al., 2003; Mouriño-Carballido et al.,
2006). Here we report with a small timescale (11 days) that in a CE, states of little to high
autotrophy occurred. Thus, phytoplankton dynamic and associated bacterial responses within
eddies not only change with time but also through space. This urges the need for more high-
resolution eddy studies in order to better estimate their impact on plankton metabolic activities
and carbon cycling.





## Conclusion

Our results highlight the ability of a CE to be an autotrophic vector towards the open ocean with organic matter freshly produced by the phytoplankton community inside. Yet, despite the strong autotrophy associated with the CE, phytoplankton exudation of DOM was not always enough to compensate for bacterial metabolic needs. Even if BP was enhanced in the CE, the BGE was low and varied substantially. This implies that heterotrophic bacteria recycle allochtonous DOM transported by the eddy and/or have issues to degrade phytoplankton DOM. Microbial metabolic activities dynamic within eddies are complex and require further investigations to understand and unravel the carbon cycling.

## Data availability

All data will be made available at the PANGEA database (data manager, webmaster: Hela Mehrtens)

## Author contribution

QD, KWB and AE designed the scientific study, analyzed the data and wrote the paper. AB, did the eddy reconstruction and both AE and JH commented on the paper.

## Competing interests:

The authors declare that they have no conflict of interest.

## Acknowledgments

We thank the captain and the crew of the *R/V Meteor* for their support during the M156 cruise. We thank J. Roa, T. Klüver and L. Scheidemann for sampling on board. We thank J. Roa and S. Golde additionally for the analysis of dissolved organic matter and T. Klüver for cell counting, bacterial and phytoplankton activities analyses. We thank B. Domeyer and R.





Suhrberg for the nutrient analyses. This study has been conducted using E.U. Copernicus
Marine Service Information. The results contain modified Copernicus Climate Change Service
information 2020. Neither the European Commission nor ECMWF is responsible for any use
that may be made of the Copernicus information or data it contains. This study is a contribution
of the REEBUS project (Role of Eddies in the Carbon Pump of Eastern Boundary Upwelling
Systems) sub-projects WP1 and WP4, funded by the BMBF (funding reference no. 03F0815A).

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
