# Peer review of "Eddy-enhanced primary production sustains heterotrophic microbial"

_Biogeosciences, 2022_

## Author Comment (AC1)

**Response to comments from reviewer #1**

The authors present data from a detailed oceanographic survey in the Eastern Tropical North Atlantic, along a transect crossing a cyclonic eddy (CE). It is a highly valuable piece of data that contribute to better understand the impact of CE on microbial metabolism. Yet, the spatial resolution for some of the biological variables is limited (primary production, bacterial production and community respiration) and some of the metabolic rates were estimated (bacterial respiration) or measured at lower temperature than in situ temperature (bacterial production and community respiration), which may be distorting the relationship among variables. It is very difficult to understand why bacterial production or respiration could not be incubated at in situ, while Pp was incubated at in situ temperature. This is a major drawback, as the method used for BP estimation actually provides exactly the same BP at 22ºC than at 14ºC, which seems rather unlikely, at least in the absence of resource limitation. This definitely requires further explanation, or even using a different model for BP estimates. Therefore, the manuscript need a major revision to clarify and, eventually, reanalyze the results. The discussion should also be accordingly revised, and avoid repeating results or speculative statements. Also, the stations should be clearly identified in all the figures, and some figures should be revised. The English usage should be also carefully revised.

Thank you for the thorough review and support of our manuscript. We agree with the reviewer that it is favorable to incubate at in-situ temperatures. For technical constraints and due to the tight sampling schedule during the cruise, we have not been able to conduct the respiration measurements at in-situ temperatures; this would have required adaptation to a large number of different temperatures. In particular within the eddy vertical temperature profiles were very variable (Fig. 1). For example, the CTD was at times deployed every few hours, but incubation times were 36 h for CR measurements with the optodes. Additionally, only one incubator (fridge) with a temperature range between 4 and 16 °C was available to us. Bacterial processes are more susceptible to temperature than primary production. In order to obtain comparable results for BP, we incubated the samples at the same temperature as the samples for CR. The well-known relationship between temperature and these rates were then used to correct for the temperature difference.

Thank you for pointing us to the unrealistic BP rates after the temperature correction. We identified a mistake in the calculations and have corrected BP and BGE estimates. Temperature-corrected values for BP at in-situ temperature (22 °C) changed by a factor of 1.9 and are thus now 1.9 times higher than the values based on the incubations at 14 °C using the equation of López-Urrutia and Morán (2007). We will change the text, Figures, and Tables accordingly.

We will revise the entire manuscript to avoid simple mistakes and to use a more standard English.

Specific comments:

Title: I suggest changing the title, as the authors do not provide growth data and also the term "accelerates" is rather confusing. Moreover, primary production appears to be enhanced in a frontal zone, not in the CE, and this should be clear already in the title.

We agree with the referee that measured rates were particularly high at the eddy front. However, we would like to keep the title a bit more general, as future studies need to demonstrate whether frontogenesis is indeed the cause for enhancing PP. We will change the title to: "Eddy-enhanced primary production  sustains microbial activities in the Eastern Tropical North Atlantic" This will include effects of the eddy on frontogenesis.

Abstract:

Line 19: revise "Mauretania" throughout the text and change to "Mauritania".

We will make the correction according to the referee's suggestion.

Line 21: revise the use of the term "cascading", which implies a temporal dimension, that has not been adequately addressed here.

We will change the term to "coupled".

Line 26: revise the use of the term parameter, which is not equivalent to the term variable. As an example, chl-a concentration is not a parameter.

We will use the term "variables" instead of "parameters"

Lines 25-27: please be more specific, and clearly indicate that the maximum concentration of phytoplankton occurred in the frontal zone.

The maximum Chl-a concentration was measured within the boundary of the eddy, while, for example, PP was highest in the frontal zone (see Fig. 2). The chosen interpolation betw\een data points might have been misleading and will be slightly adjusted to make that clearer.

Line 36: indicate to what this percentage is referred to.

The percentage refers to % of $PP_{DOC}$ and we will indicate this.

Lines 36-37: I do not think that PP/BCD reflects the metabolic state of the microbial community. I suggest either using PP/CR as an estimation of the metabolic state of the microbial plankton community, or use BCD/PP as indication of the fraction of PP production that is processed by bacteria. Please be specific. A PP/BCD>1 does not necessarily imply an autotrophic balance.

We thank the reviewer for this valuable comment. We will follow the suggestion to use PP/CR as an estimation of the metabolic state of the microbial plankton community.

Introduction:

Line 90: indicate that you refer to bacterial biomass production.

We will make the correction according to the referee's suggestion.

Line 92: provide more recent references for the effect of DOM on BGE.

We will make the correction according to the referee's suggestion.

Line 100: provide references for BR on eddies.

We will add references for BR.

Lines 105-107: this part is somehow repetitive with information in lines 79-82. Please revise and avoid repetition.

Lines 79-82 will be merged with those in lines 105-107 of the original manscript to avoid repetition. Proposed new text:

*"Yet, insight into the distribution of phytoplankton and their activities within mesoscale eddies is limited due to a lack of sufficient fine-scale vertical and horizontal resolution studies to adequately describe these distributions."*

Lines 109-111: this part is also repetitive with that in lines 69-71. Please revise.

Lines 69-71 will be removed to avoid repetition.

Line 116: please specify the spatial resolution of the study.

We will specify the spatial resolution of the study:

*"We studied the impact of a CE on microbial carbon cycling along a 900 km zonal corridor"*

Materials and methods:

Lines 133-134: please clarify what you mean by "consecutive optimized identification of the eddy".

We will change the text to:

*"[…], which made it difficult to identify the center of the eddy and required rerouting of the ship's track during the survey."*

Lines 141-144: please provide also information about the temporal sequencing of the survey. A supplementary table indicating the sampling sate of each stations would be nice.

The sampling date and time will be added to Supplementary Table 1.

Lines 158-162: please re-write for clarity and English usage.

We will revise the sentence to read:

"Just beyond of the eddy periphery, at St. E3, a front was observed with surface temperature and salinity (not compensated by density) clearly different from the adjacent stations (Fig. 1b)."

Figure 1: The cruise track is not visible in the figure. Else, the positions of the stations in the CE are not fully visible, I suggest making a different graph for the stations within the CE. Finally, increase the suze of the symbols in plots b, c and d.

We will revise the caption by replacing "Cruise track" with "Sampling stations" and add a zoom-in showing the stations within the eddy. The size of symbols will be increased in panels (b), (c) and (d). Please see below for a draft of these changes.

[Figure]

Line178: nitrate and nitrite lack the symbol of the charge

Will be corrected.

Lines 179-181: please provide a reference for this statement.

We will add the following reference:

Carlson, C. A.: Production and Removal Processes. Chapter 4 in Biogeochemistry of Marine Dissolved Organic Matter, Editor(s): Hansell D. A., Carlson, C. A. AP, 805, 91–151. https://doi.org/10.1016/b978-012323841-2/50006-3. 2002.

Lines 187 and 196: please clarify if you measured dAA or dHAA.

We measured dHAA and will add this information accordingly.

Lines218-221: this is not equivalent to autotrophic plankton biomass, as it is including only pico and nanoplankton. Please use a term that clearly states this to avoid any possible confusion.

We will change the term to "autotrophic pico-and nanoplankton biomass".

Line 227: two duplicate samples and only one killed control is not really sufficient to get accurate estimates of BP.

Due to time, equipment capacity, and workload during the expedition we were only able to collect and analyse duplicates for BP. Yet, we entrust our data as the standard deviation for each sample were relatively low (see figure below for individual measurements of the duplicates).

[Figure]

Figure: Individual data points of duplicate measurements of bacterial biomass production (BP) rates from 0-200 m depth from all samples.

Line 229: the authors should clearly explain why they did not measure BP at in situ temperature. This is a major limitation of their work, and is not sufficiently justified.

We agree with the reviewer that incubations at in situ temperature would have been favorable. However, due to technical issue, we could only measured CR in an incubator (fridge) with a temperature range of 4 to 16 °C.Both BP and CR rates are temperature dependent (e.g., López-Urrutia and Morán, 2007; Regaudie-de-Gioux et al., 2012, Yvon-Durocher et al., 2012) and in order to compare them, we decided to incubate them at the same temperature (14 °C).

Line 233: the author should consider to use a different model to estimate BP at 22°C form estimates at 14°C.

We thank the referee for pointing us to this. We found a mistake in our calculations which will be corrected and now provides a more realistic conversion factor of the BP rates. All results and figures will be corrected accordingly. Please see our response to your general comment for the changes in BP rates between the original and the revised version of the manuscript.

Line 236: again, the authors should justify the reason why they did not measure CR at in situ temperature. Also, they should explain why they conducted incubations > 24 h, and when. Finally, the number of replicates for CR should be also indicated.

Please see the explanation above concerning the incubation temperature.

For the incubation time, we measured the decrease of oxygen at several time points (0 h, 6 h, 12 h, 24 h and 36 h). The relatively long incubation time was chosen due to the low CR typically observed in oligotrophic water (e.g Reinthaler et al., 2006) and to be consisent throughout the cruise. The method is fully described in the supplementary information.

We will included the information that CR was measured in quadruplicate.

Lines 253-254: again, the number of replicates is too low.

While we understand that more replicates for PP rates would have been favorable, during sea-going expeditions, sample material and time is often limiting, restricting the number of replicates that can be analyzed. Our duplicate analysis showed highly similar results (see figure showing individual data points for each sample). Additionally, we have measured and published PP data from many oceanic regions with variable number of replicates and the obtained rates have always shown highly similar similar results. Thus, we believe that our PP data provide accurate estimates.

[Figure]

Figure: Individual data points of duplicate measurements of total primary production (PP_TOT) from all samples. When only one data point is visible, data point are overlying.

Line 254: indicate where PP incubations were done (controlled chamber?).

We will add the information that the incubations were done in an incubator.

Line 264: the author should justify the use of 0.4 instead of 0.2 microns PC filters to separate the dissolved fraction. Some bacteria can pass through 0.4 microns.

We agree with the reviewer that some bacteria might pass through 0.4 µm filters. However, filtration at 0.4 µm was initially (in the older literature) selected because it corresponds roughly to the upper limit size of viruses and the lower limit of bacteria. Comparisons of results from studies using different filter pore-sizes is highly uncertain. Thus, we decided to stick to the convetional method based on 0.4 µm filters.

Lines 269-270: please re-write for clarity.

We will revise the sentence. Proposed new text:

"To determine $PP_{DOC}$, 4 mL of filtrate were transferred to 20 mL scintillation vials and acidified with 100 µL 1N HCl. Scintillation vials were left open in the fume hood for 14 hours to remove inorganic carbon."

Line302-303: clarify the method of integration. Is the same as the trapezoid rule?

The midpoint rule approximates the definite integral using rectangular regions whereas the trapezoidal rule approximates the definite integral using trapezoidal approximations. We preferred to use the midpoint rule as it provides more accurate integrations especially when only three data points are used as for the 100 m integration that we did.

Results:

Figure 2: please clearly indicate in the plots the identification code of each station. Also increase the size of the dots.

We will increase the size of the dots and include all the station names.

Line 373: clarify that this is not autotrophic plankton biomass, it is only pico and nanoplankton using another term to refer to this.

We will change the term to "autotrophic pico- and nanoplankton biomass"

Lines 379-381: as stated above, it is better using a term that clearly define what this variable is, and thus, this sentence can be removed.

We will remove this sentence.

Table 1: please clarify how do you integrate down to 100 m in stations lacking samples below 50-75 m. Revise the use of the term "parameter". Why do the authors specify depths and sampling date for only some of the stations?

For all stations, samples for most parameters exist down to 200 m. The only exeptions are PP and CR. For the extrapolation, the shallowest value was extrapolated to 0 m and then the midpoint rule was applied down to 100 m. For PP only the top three depths were sampled. The fourth depth corresponded to the base of the photic zone based on Chl fluorescence profiles. This depth was extrapolated to a value of zero. The same was applied for CR. We will remove the information about the sampling depth and time point from the caption as they are disturbing here. Instead, we will provide this information in the SI Table 1 and will refer to the table in the caption of Table 1.

Table1: the differences between integrated chla- between EDZ1 and E3 are weird and not expected form what is presented in figure 3 (although in figure 3 the stations are not clearly indicated). Overall the results section is very difficult to follow due to the lack of station labels in figures.

We will add station labels to the Figures.

Figure 3: please add station labels and increase the size of dots. Rename the variable AutPI for clarity (it is only pico and nano plankton biomass).

We will make the change according to the referee's suggestion.

Line 433: clarify is you refere to integrated or volumetric BP rates

We refered to volumetric BP rates and will include this information in the revised verion.

Line 430: PP/BCD < 1, does not indicate heterotrophic balance or conditions, it just indicates that concurrent PP is not fulfilling BCD. Revise and be more specific. I suggest either using PP/CR as an estimation of the metabolic state of the microbial plankton community, or use BCD/PP as indication of the fraction of PP production that is processed by bacteria.

We will make the correction according to the referee's suggestion and use $PP_{TOT}/CR$ as an estimation of the metabolic state of the microbial plankton community.

Line 453: I suggest including this calculation ($PP_{DOC}/BCD$) in table 2.

We will include the suggested calculation in Table 2

Figure 4: please add station labels and increase the size of dots. Clearly state in the figure legend that BP and BR are stimates and indicate the method used for that estimation.

We will add the station labels, increase the size of the dots and change the legend to:

"CR and BP rates at in-situ temperature were estimated based on López-Urrutia and Morán (2007). BR rates were estimated from measured and temperature-corrected CR rates based on Regaudie-de-Gioux and Duarte (2012). Details are provided in the methods section and the SI."

Table 2: I suggest using PP/CR and BCD/PP as more insightful ratios than BCD/PP.

As mentioned above, we will follow the reviewer's suggestion.

Line 484: revise the usage of the term "indices" here, as it does not reflect the content of the section.

We will remove that section including the figures from the manuscript as it seems unnecessary.

Line 488: the correlation between cell-specific BR and BGE is spurious as both contain the variable BR. Please remove form the analysis.

See above.

Figure 5: I suggest representing BCD/PP, I find it more intuitive that the inverse.

We will use $PP_{TOT}$/CR as suggested above by the reviewer to estimate the metabolic state of the microbial plankton community.

Lines 491-493: I suggest removing also the correlation between chl-a and the biomass of pico and nanophytoplankton, as it is not necessary. It is enough indicating that the discrepancies are due to the fact that chl-a is total, and the biomass is only form small phytoplankton.

We will remove this section including the plot from the manuscript.

Figure 6: I suggest removing plot (a) (because it is spurious) and (b) (as iti is not necessary). Maybe the authors could add plots relating Chl-a vs. BP and/or BCD vs. $PP_{DOC}$.

See above.

Lines 513-524 and figure 7: please revise to eliminate the spurious correlations (e.g. BCD vs BR or BR; PPtot vs PPdoc). The authors could try to calculate correlation using data not affected and affected by the CE.

We will follow the referee's suggestion and make two correlation matrices using data not affected and affected by the CE (see below for proposed new figure). We will change the text accordingly and also remove the spurious correlations.

[Figure]

**(a)** Open ocean + Coastal

[Figure]

**(b)** Cyclonic eddy + Frontal zone

Correlation coefficient (r)

Figure: Correlations of biochemical parameters, metabolic activities, and bacterial abundance in the upper 100 m in (a), the transect excluding eddy-influenced samples, (i.e., coastal and open ocean stations) and (b) the eddy influenced samples. Statistical significance: '***'< 0.001, '**'< 0.01, '*'< 0.05.

Discussion:

Lines 545-546: this is speculative as the authors do not have data about the fraction of large phytoplankton. The relation between chl-a and biomass are also affected by factors such as photoacclimation. The authors can only guess that in more productive stations large phytoplankton is likely more relevant, but they do not have data to support that statement.

We will follow the referee's suggestion and remove the statement.

Line 553-555: this is again speculative, the authors do not have date on the contribution of small planktons, they only have total chl-a and the biomass of the small fraction, but the relation between chl-a and biomass is not straightforward. I suggest eliminating this statement

We will follow the referee's suggestion and remove the statement.

Lines 565: I suggest using an alternative to "compression", such as e.g. "uplifting".

We will make the correction according to the referee's suggestion.

Lines 567-568: revise for English usage.

We will revised this sentence to read:

"Similar uplifting of Chl-a isolines towards the surface have been reported for other eddies (Lochte and Pfannkuche 1987; Feng et al., 2007; Noyon et al., 2019) and might result from phytoplankton relocation through intense vertical mixing by strong surface winds (Feng et al., 2007; Noyon et al., 2019)."

Lines 573-575: revise the sentence, it is hard to follow the reasoning.

We will revise this sentence and integrate the statement into the previous ones. Proposed new text:

"Similar uplifting of Chl-a isolines towards the surface have been reported in other eddies (Lochte and Pfannkuche 1987; Feng et al., 2007; Noyon et al., 2019) and have been suggested to rusult from phytoplankton relocation through intense vertical mixing from strong surface winds (Feng et al., 2007; Noyon et al., 2019). Before our eddy survey, strong surface winds occurred offshore (SI Fig. S7), which might explain the high Chl-a concentration (>0.5 µg L-1) that we found at the surface (5 m) in all stations within the CE."

Lines 576-580: delete as this is mostly results.

We will delete this statement.

Lines 594-598: again very speculative. The authors do not have data about the presence of diatoms or dinoflagellates in this study. Delete or rewrite.

We will delete this statement.

Line 603: the use of the term "diversity" is not appropriate here, as the authors only provide data of a couple of functional phytoplankton groups.

We will avoid the term "diversity".

Lines 603-606: revise English usage as it is very difficult to understand the sentence.

We will revise the sentence. Proposed new text:

"Our flow cytometry data (SI Fig. S7) showed that Cyanobacteria (Synechococcus) and eukaryotic pico- and nanoplankton within the CE were unevenly distributed. This, suggest that the phytoplankton community of the CE was likely distinct from the surrounding waters, but also variable on the submesoscale within the CE. This is consistent with previous studies on phytoplankton distributions in eddies (e.g., Lochte and Pfannkuche, 1987; Lasternas et al., 2013; Hernández-Hernández et al., 2020)."

Lines 607-608: delete this sentence, the authors do not have data on phytoplankton taxonomy, only flow cytometry counts of different groups based on scatter and fluorescence.

This sentence will be deleted.

Lines 619-621: this sentence is just repeating results. Please, delete.

This sentence will be deleted.

Line 621-625: please revise English usage.

The sentence will be revised. Proposed new text:

"We emit two hypotheses regarding this distribution: 1) the lower PER was due to a higher proportion of larger phytoplankton (e.g., diatoms) which have lower turnover rates and therefore lower PER and/or 2) the upwelling of nutrients generated by the CE might have enhanced the physiological health of the phytoplankton community (Agustí and Duarte, 2013; Laternas and Agustí, 2014)."

Lines 630-631: please indicate where this correlation is found in the results, as the correlation matrix in figure 7 was calculated including all data.

We will divide original Figure 7 into two correlation matrices as suggested by the reviewer in an earlier comment and add the reference to the figure in the main text.

Lines 631-633: revise English usage. Else, it is hard to see such continuous trends in HB or PP in the figures.

We will change the sentence to:

"Along the zonal transect, in the stations not affected by the eddy (open ocean+coastal stations), a significant positive correlation between HB abundance and PP$_{TOT}$ rates was observed (Fig. 6A)."

Line 635: delete this first sentence.

Will be deleted.

Line 636: explain the acronym CanUS.

We will explain the acronym.

Lines639-641: re-write for clarity. Again, avoid statements about phytoplankton compositions, as the authors are not reporting such data (they only have cytometric groups).

We will remove this statement.

Line 656: please town down, change "state" to "suggest".

We will make the change according to the referee's suggestion

Lines 661-662: certainly BGEs are very low, which may be partially related to a severe underestimation of BP (see general comments and comments to the materials and methods section).

As outlined above, BP rates and BGEs will be corrected due to a mistake in the calculations.

Lines 686-688: please delete references tp the presence of diatoms and/or dinofñagellates as these data are not provided. Also town donw the statement.

We will change the sentence to:

"As stated previously, the upwelling induced by the CE and the Frontal Zone led to higher phytoplankton biomass, which was likely responsible for this increase in BP."

Lines 689-706: all this discussion must be revised once BP estimates are clarified. Also engñish usage should be revised.

The paragraph will be revised after the correction as outlined above.

Lines 707-714: all this paragraph is about an spurious relationship. In addition, the authors do not have data on bacterial community compostion. I suggest deleting it.

We will delete this part of the discussion according to the referee's suggestion.

Line 715: revise the usage of the term "growth" as this variable was not included in this study.

We will revise the use of the term "growth". Proposed new text:

"Our results show that BGE is not proportional to the amount of DOM received through exudation but rather depends on the different requirements between respiration and biomass production."

Lines 720-722: revise as it is very difficult to follow the reasoning, as phytoplankton taxonomic composition is not provided in this study.

We will change the sentence to:

"Here we hypothesise that in CEs, which cross oligotrophic waters in the ETNA, BGE variability depends on both BP through phytoplankton biomass and BR through the amount and quality of SL-DOC."

Lines 732-736: revise English usage. In addition, revise statements about temporal dynamics, which does not seem to be adequately resolved in this survey.

We will change the sentences to:

"Here we showed that both autotrophy and heterotrophy can occur at the same time within a single eddy. This urges the need for more high-resolution eddy studies in order to better estimate their impact on plankton metabolic activities and carbon cycling."

---

## Author Comment (AC2)

**Response to comments from reviewer #2**

General comments:

This study mainly investigated how cyclonic eddy (CE) affects heterotrophic bacterial activities in the surface waters of the eastern tropical North Atlantic by using measurements of various parameters related to the microbial activities. The measurements are valuable for understanding the effect of CE on the microbial activities. The study is interesting and suitable for the scope of this journal. However, there are several points which should be made clearer before publication. Please find below specific comments.

Major comments:

In this study, bacterial biomass production (BP) and community respiration (CR) rates are the most important parameters. Those rates depend on in situ temperature. However, BP and CR were estimated not at in situ temperature but at 14 ℃. The reason why the authors used 14 ℃ as incubation temperature should be mentioned.

We thank the reviewer for the thorough review of our manuscript. We agree with the reviewer that incubating at in situ temperatures would have been favorable. As explained as response to a comment from reviewer #1 regarding the same criticism, we had to choose a different incubation temperature than in situ temperature for CR for technical reasons. Only a fridge with a temperature range between 4 °C to 16 °C was available to us that could be used with the optode setup. To obtain comparable results for BP and CR, we have used the same temperature for both rate measurements, i.e., 14 °C. The well-documented dependence of CR and BP rates on temperature (e.g., López-Urrutia and Morán, 2007; Yvon-Durocher et al., 2012) allowed us to correct for the difference between incubation and in situ temperature. The temperature correction is explained in detail in the Methods section and the SI.

There are several points that are not based on the clear evidences:

1)bacterial respiration rates are related to semi-labile (SL) dissolved organic carbon (DOC) concentration (lines 651-652), 2) microbes in the CE preferentially use SL-DOC (lines 696-697), 3) microbes do not grow in tandem with the increase in dissolved primary production (PP$_{DOC}$) but are related to the different requirement between BR and BP (lines 715-717), and 4) bacterial growth efficiency (BGE) varies depending on both BP via phytoplankton taxonomical composition and BR via the quantity and quality of the SL-DOC (lines 720-722). The statements 1), 2) and 3) are probably based on the results of correlations between relevant parameters (Fig. 7), while the statement of 4) is probably based on Table 2, Figs. 6a,b and 7. The results that each statement is based on

are not clear at present. Please make the statements clearer by referring to proper results.

The reviewer is correct. The statements 1-3 are based on Fig. 7 (Fig.6 in the original manuscript), while statement 4 is based on Fig. 7 (Fig.6 in the original manuscript) and Table 2. Some of the statements would be changed in a revised version due to comments from reviewer #1, but where appropriate, we will refer to the Figure or Table for each of the statements.

Specific comments:

Line 271: How long scintillation vials are left open after addition of HCl should be described and proper reference should also be added here. I wonder if all dissolved inorganic carbon can be removed by the method or not.

The scintillation vials were left open for 14 hours after addition of HCL according to the method described in Steemann Nielsen (1952). We will add this information and the reference. The blanks showed no evidence of remaining inorganic carbon.

Figure 3: Adding the depth profiles of BGE and $PP_{DOC}$ is helpful for readers.

We will add the depth profiles to the Figures 3 and 4, respectively.

Lines 569 and 593: The authors mentioned high vertical mixing due to strong surface winds. Showing the strong surface wind data would be helpful for readers.

The wind data was shown in Supplementary Figure 6 of the original submission. We will refer to this Figure.

Lines 608-609: Mixed layer depths should be added to Figures 2, 3, and 4 for easy readability.

We will consider the reviewer's suggestion. However, after including the mixed layer depth in the plots, we feel that the figures might be overloaded. Instead, we suggest to show the mixed layer depth in a new Figure (3a, see below) and in Table S1.

[Figure]

Proposed new Figure 3: Spatial distribution of the mixed layer depth (a) and chlorophyll *a* integrated over the upper 100 m depth (b) during M156.

Lines 630-631: Please clarify whether all the data of HB abundance and particulate primary production or a part of those data were used

The statement was based on the correlation of the parameters in the open ocean and coastal stations only (stations not affected by the eddy). According to a comment from reviewer #1, we will include a correlation analysis for the eddy-influenced stations and the stations not influenced by the eddy in two panels A and B (see proposed new figure below) and remove the original correlation analysis. We will refer to the new Figure and change the text to make it clearer that the statement is based on the correlations for the stations not affected by the eddy.

[Figure]

**(a)** Open ocean + Coastal

[Figure]

**(b)** Cyclonic eddy + Frontal zone

Correlation coefficient (r)

Proposed new Figure 7: Pearson correlation matrix of biochemical parameters, metabolic activities, and bacterial abundance in the upper 100 m in samples not influenced by the cyclonic eddy (i.e., coastal and open ocean stations) (a) and samples influenced by the cyclonic eddy (b). Statistical significance: '***'< 0.001, '**'< 0.01, '*'< 0.05.

---

## Author Response (AR1)

**Response to comments from reviewer #1**

The authors present data from a detailed oceanographic survey in the Eastern Tropical North Atlantic, along a transect crossing a cyclonic eddy (CE). It is a highly valuable piece of data that contribute to better understand the impact of CE on microbial metabolism. Yet, the spatial resolution for some of the biological variables is limited (primary production, bacterial production and community respiration) and some of the metabolic rates were estimated (bacterial respiration) or measured at lower temperature than in situ temperature (bacterial production and community respiration), which may be distorting the relationship among variables. It is very difficult to understand why bacterial production or respiration could not be incubated at in situ, while Pp was incubated at in situ temperature. This is a major drawback, as the method used for BP estimation actually provides exactly the same BP at 22ºC than at 14ºC, which seems rather unlikely, at least in the absence of resource limitation. This definitely requires further explanation, or even using a different model for BP estimates. Therefore, the manuscript need a major revision to clarify and, eventually, reanalyze the results. The discussion should also be accordingly revised, and avoid repeating results or speculative statements. Also, the stations should be clearly identified in all the figures, and some figures should be revised. The English usage should be also carefully revised.

Thank you for the thorough review and support of our manuscript. We agree that it would have been favorable to incubate at in-situ temperatures. Due to technical constraints and to the tight sampling schedule during the cruise, we have not been able to conduct the incubation measurements (BP and CR) at in-situ temperatures; this would have required adaptation to a large number of different temperatures. In particular within the eddy, vertical temperature profiles were highly variable (see Fig. 1 of original submission). We collected and analysed water from the surface down to 800 m depth and because of the technical and time constraints, an average temperature was chosen (14°C). However, since effects of eddies on biogeochemical processes is highest in the sunlit surface, we only analyzed the samples from the top 100 m in this study. PP was incubated at 22°C because this was the average in situ temperature in the photic zone at almost all stations and PP was only determined from these depths. . In order to obtain comparable results for BP and CR, we incubated the samples at the same temperatures. The well-known relationship between temperature and these rates were then used to correct for the temperature difference. We have added information about the chosen temperatures in the revised version.

Thank you for pointing us to the unrealistic BP rates after the temperature correction. We identified a mistake in the calculations and have corrected the BP and BGE estimates. Temperature-corrected values for BP at in-situ temperature (22 °C) changed by a factor of 1.9 and are thus now 1.9 times higher than the values based on the incubations at 14 °C using the equation of López-Urrutia and Morán (2007). We have changed the text, Figures, and Tables accordingly.

We have revised the entire manuscript to avoid simple mistakes and to use a more standard English.

Specific comments:

Title: I suggest changing the title, as the authors do not provide growth data and also the term "accelerates" is rather confusing. Moreover, primary production appears to be enhanced in a frontal zone, not in the CE, and this should be clear already in the title.

We agree that measured rates were particularly high at the eddy front. However, we would like to keep the title a bit more general, as future studies need to demonstrate whether frontogenesis is indeed the cause for the enhanced PP that we have observed. To avoid confusion with the term 'accelerates', we have changed the title to: "Eddy-enhanced primary production sustains heterotrophic microbial activities in the Eastern Tropical North Atlantic ".

Abstract:

Line 19: revise "Mauretania" throughout the text and change to "Mauritania".

We have made the correction according to the referee's suggestion.

Line 21: revise the use of the term "cascading", which implies a temporal dimension, that has not been adequately addressed here.

We have changed the term to "coupled".

Line 26: revise the use of the term parameter, which is not equivalent to the term variable. As an example, chl-a concentration is not a parameter.

We now use the term "variables" instead of "parameters" throughout the manuscript.

Lines 25-27: please be more specific, and clearly indicate that the maximum concentration of phytoplankton occurred in the frontal zone.

The maximum Chl-a concentration was measured within the boundary of the eddy, while, for example, PP was highest in the frontal zone (see Fig. 2). The chosen interpolation between data points might have been misleading and have been slightly adjusted to make that clearer (see slightly modified Fig. 4).

Line 36: indicate to what this percentage is referred to.

The percentage refers to % of PP$_{DOC}$ and we have indicated this.

Lines 36-37: I do not think that PP/BCD reflects the metabolic state of the microbial community. I suggest either using PP/CR as an estimation of the metabolic state of the microbial plankton community, or use BCD/PP as indication of the fraction of PP production that is processed by bacteria. Please be specific. A PP/BCD>1 does not necessarily imply an autotrophic balance.

Thank you for this valuable comment. We followed the suggestion to use PP/CR as an estimation of the metabolic state of the microbial plankton community.

Introduction:

Line 90: indicate that you refer to bacterial biomass production.

We have made the correction according to the referee's suggestion.

Line 92: provide more recent references for the effect of DOM on BGE.

We have made the correction according to the referee's suggestion.

Line 100: provide references for BR on eddies.

We have added references for BR.

Lines 105-107: this part is somehow repetitive with information in lines 79-82. Please revise and avoid repetition.

Lines 79-82 have been merged with those in lines 105-107 of the original manscript to avoid repetition. Proposed new text:

"Yet, insight into the distribution of phytoplankton and their activities within mesoscale eddies is limited due to a lack of sufficient fine-scale vertical and horizontal resolution studies to adequately describe these distributions." Lines 101-104

Lines 109-111: this part is also repetitive with that in lines 69-71. Please revise.

Lines 69-71 have been removed to avoid repetition.

Line 116: please specify the spatial resolution of the study.

We have specified the spatial resolution of the study:

"We studied the impact of a CE on microbial carbon cycling along a 900 km zonal corridor". Line 109.

Materials and methods:

Lines 133-134: please clarify what you mean by "consecutive optimized identification of the eddy".

We have changed the text to:

"[…], which made it challenging to identify the center of the eddy and required rerouting of the ship's track during the survey." Lines 131-133.

Lines 141-144: please provide also information about the temporal sequencing of the survey. A supplementary table indicating the sampling sate of each stations would be nice.

The sampling date and time have been added to Supplementary Table 1.

Lines 158-162: please re-write for clarity and English usage.

We have revised the sentence. It now reads:

"Just beyond of the eddy periphery, at St. E3, a front was observed with surface temperature and salinity (not compensated by density) clearly different from the adjacent stations (Fig. 1b)."

Figure 1: The cruise track is not visible in the figure. Else, the positions of the stations in the CE are not fully visible, I suggest making a different graph for the stations within the CE. Finally, increase the suze of the symbols in plots b, c and d.

We have revised the caption by replacing "Cruise track" with "Sampling stations" and added a zoom-in showing the stations within the eddy. The size of the symbols has been increased in panels (b), (c) and (d). Please see below.

[Figure]

Line178: nitrate and nitrite lack the symbol of the charge

Corrected.

Lines 179-181: please provide a reference for this statement.

We have added the following reference:

Carlson, C. A.: Production and Removal Processes. Chapter 4 in Biogeochemistry of Marine Dissolved Organic Matter, Editor(s): Hansell D. A., Carlson, C. A. AP, 805, 91–151. https://doi.org/10.1016/b978-012323841-2/50006-3. 2002.

Lines 187 and 196: please clarify if you measured dAA or dHAA.

We measured dHAA and have added this information accordingly.

Lines218-221: this is not equivalent to autotrophic plankton biomass, as it is including only pico and nanoplankton. Please use a term that clearly states this to avoid any possible confusion.

We have changed the term to "autotrophic pico-and nanoplankton biomass".

Line 227: two duplicate samples and only one killed control is not really sufficient to get accurate estimates of BP.

Due to time, equipment capacity, and workload during the expedition, we were only able to collect and analyse duplicates for BP. Yet, we entrust our data as the standard deviation for each sample was relatively low (see figure below for individual measurements of the duplicates).

[Figure]

Figure: Individual data points of duplicate measurements of bacterial biomass production (BP) rates from 0-200 m depth from all samples.

Line 229: the authors should clearly explain why they did not measure BP at in situ temperature. This is a major limitation of their work, and is not sufficiently justified.

Please see our response to your general comment.

Line 233: the author should consider to use a different model to estimate BP at 22ºC form estimates at 14ºC.

Thank you for pointing us to this. We found a mistake in our calculations, which have been corrected and now provide a more realistic conversion factor of the BP rates. All results and figures have been corrected accordingly. Please see our response to your general comment for the changes in BP rates between the original and the revised version of the manuscript.

Line 236: again, the authors should justify the reason why they did not measure CR at in situ temperature. Also, they should explain why they conducted incubations > 24 h, and when. Finally, the number of replicates for CR should be also indicated.

Please see the response to the general comment regarding the incubation temperature.

For the incubation time, we measured the decrease of oxygen at several time points (0 h, 6 h, 12 h, 24 h and 36 h). The relatively long incubation time was chosen due to the low CR typically observed in oligotrophic water (e.g Reinthaler et al., 2006) and to be consisent throughout the cruise. We started with 24 h and then we added an additional time point (36 h) due to the low respiration rates (see example below).

[Figure]

We have included the information that CR was measured in quadruplicate.

Lines 253-254: again, the number of replicates is too low.

While we understand that more replicates for PP rates would have been favorable, during sea-going expeditions, sample material and time is often limiting, restricting the number of replicates that can be analyzed. Our duplicate analysis showed highly similar results (see figure showing individual data points for each sample below). Additionally, we have measured and published PP data from many oceanic regions with variable number of replicates and the obtained rates have always shown highly similar similar results. Thus, we believe that our PP data provide accurate estimates.

[Figure]

Figure: Individual data points of duplicate measurements of total primary production (PP$_{TOT}$) from all samples. When only one data point is visible, data point are overlying.

Line 254: indicate where PP incubations were done (controlled chamber?).

We added the information that the incubations were done in an incubator. Line 261.

Line 264: the author should justify the use of 0.4 instead of 0.2 microns PC filters to separate the dissolved fraction. Some bacteria can pass through 0.4 microns.

We agree with the reviewer that some bacteria might pass through 0.4 μm filters. However, filtration at 0.4 μm was initially (in the older literature) selected because it corresponds roughly to the upper limit size of viruses and the lower limit of bacteria. Comparisons of results from studies using different filter pore-sizes is highly uncertain. Thus, we decided to stick to the convetional method based on 0.4 μm filters.

Lines 269-270: please re-write for clarity.

We revised the sentence. It now reads:

"To determine $PP_{DOC}$, 4 mL of filtrate were transferred to 20 mL scintillation vials and acidified with 100 μL 1N HCl. Scintillation vials were left open in the fume hood for 14 hours to remove inorganic carbon." Lines 276-278.

Line302-303: clarify the method of integration. Is the same as the trapezoid rule?

The midpoint rule approximates the definite integral using rectangular regions whereas the trapezoidal rule approximates the definite integral using trapezoidal approximations. We preferred to use the midpoint rule as it provides more accurate integrations especially when only three data points are used as for the 100 m integration that we did.

Results:

Figure 2: please clearly indicate in the plots the identification code of each station. Also increase the size of the dots.

We have increased the size of the dots and included all the station names.

Line 373: clarify that this is not autotrophic plankton biomass, it is only pico and nanoplankton using another term to refer to this.

We have changed the term to "autotrophic pico- and nanoplankton biomass"

Lines 379-381: as stated above, it is better using a term that clearly define what this variable is, and thus, this sentence can be removed.

We have removed this sentence.

Table 1: please clarify how do you integrate down to 100 m in stations lacking samples below 50-75 m. Revise the use of the term "parameter". Why do the authors specify depths and sampling date for only some of the stations?

For all stations, samples for most parameters exist down to 200 m. The only exceptions are PP and CR. For the extrapolation, the shallowest value was extrapolated to 0 m and then the midpoint rule was applied down to 100 m. For PP only the top three depths were sampled. The fourth depth corresponded to the base of the photic zone based on Chl fluorescence profiles. This depth was assumed to be zero. The same was applied for CR. We removed the information about the sampling depth and time from the caption as they seem to be disturbing here. Instead, we now provide this information in the SI Table 1 and refer to the table in the caption of Table 1.

Table1: the differences between integrated chla- between EDZ1 and E3 are weird and not expected form what is presented in figure 3 (although in figure 3 the stations are not clearly indicated). Overall the results section is very difficult to follow due to the lack of station labels in figures.

We have added station labels to the Figures and revised the description of the results. We hope it is easier to follow now.

Figure 3: please add station labels and increase the size of dots. Rename the variable AutPI for clarity (it is only pico and nano plankton biomass).

We hve made the change according to the referee's suggestion.

Line 433: clarify is you refere to integrated or volumetric BP rates

We refered to volumetric BP rates and have included this information in the revised verion.

Line 430: PP/BCD < 1, does not indicate heterotrophic balance or conditions, it just indicates that concurrent PP is not fulfilling BCD. Revise and be more specific. I suggest either using PP/CR as an estimation of the metabolic state of the microbial plankton community, or use BCD/PP as indication of the fraction of PP production that is processed by bacteria.

We have made the correction according to the referee's suggestion and use $PP_{TOT}/CR$ as an estimation of the metabolic state of the microbial plankton community.

Line 453: I suggest including this calculation ($PP_{DOC}/BCD$) in table 2.

We have included the suggested calculation in Table 2

Figure 4: please add station labels and increase the size of dots. Clearly state in the figure legend that BP and BR are stimates and indicate the method used for that estimation.

We have added the station labels, increased the size of the dots and changed the legend to:

"BP and CR rates at in-situ temperature were estimated based on López-Urrutia and Morán (2007) and on Regaudie-de-Gioux and Duarte (2012). BR rates were estimated from measured and temperature-corrected CR rates based on Aranguren-Gassis et al, (2012). Details are provided in the methods section and the SI." Lines 475-478.

Table 2: I suggest using PP/CR and BCD/PP as more insightful ratios than BCD/PP.

As mentioned above, we followed the reviewer's suggestion.

Line 484: revise the usage of the term "indices" here, as it does not reflect the content of the section.

We removed that section including the figures from the manuscript as it seems unnecessary.

Line 488: the correlation between cell-specific BR and BGE is spurious as both contain the variable BR. Please remove form the analysis.

See above.

Figure 5: I suggest representing BCD/PP, I find it more intuitive that the inverse.

We now use $PP_{TOT}/CR$ as suggested above by the reviewer to estimate the metabolic state of the microbial plankton community.

Lines 491-493: I suggest removing also the correlation between chl-a and the biomass of pico and nanophytoplankton, as it is not necessary. It is enough indicating that the discrepancies are due to the fact that chl-a is total, and the biomass is only form small phytoplankton.

We have removed this section including the plot from the manuscript.

Figure 6: I suggest removing plot (a) (because it is spurious) and (b) (as iti is not necessary). Maybe the authors could add plots relating Chl-a vs. BP and/or BCD vs. $PP_{DOC}$.

See above.

Lines 513-524 and figure 7: please revise to eliminate the spurious correlations (e.g. BCD vs BR or BR; PPtot vs PPdoc). The authors could try to calculate correlation using data not affected and affected by the CE.

We followed the referee's suggestion and havce made two correlation matrices using data not affected and affected by the CE (see below for proposed new figure). We changed the text accordingly and also removed the spurious correlations.

[Figure]

**(a)**

Open ocean
+ Coastal

[Figure]

**(b)**

Cyclonic eddy
+ Frontal zone

Correlation coefficient (r)

Figure: Correlations of biochemical parameters, metabolic activities, and bacterial abundance in the upper 100 m in (a), the transect excluding eddy-influenced samples, (i.e., coastal and open ocean stations) and (b) the eddy influenced samples. Statistical significance: '***'< 0.001, '**'< 0.01, '*'< 0.05.

Discussion:

Lines 545-546: this is speculative as the authors do not have data about the fraction of large phytoplankton. The relation between chl-a and biomass are also affected by factors such as photoacclimation. The authors can only guess that in more productive stations large phytoplankton is likely more relevant, but they do not have data to support that statement.

We followed the referee's suggestion and have removed the statement.

Line 553-555: this is again speculative, the authors do not have date on the contribution of small planktons, they only have total chl-a and the biomass of the small fraction, but the relation between chl-a and biomass is not straightforward. I suggest eliminating this statement

We followed the referee's suggestion and have removed the statement.

Lines 565: I suggest using an alternative to "compression", such as e.g. "uplifting".

We have made the correction according to the referee's suggestion.

Lines 567-568: revise for English usage.

We have revised this sentence to read:

"Similar uplifting of Chl-a isolines towards the surface have been reported for other eddies (Lochte and Pfannkuche 1987; Feng et al., 2007; Noyon et al., 2019) and might result from phytoplankton relocation through intense vertical mixing by strong surface winds (Feng et al., 2007; Noyon et al., 2019)." Lines 569-572

Lines 573-575: revise the sentence, it is hard to follow the reasoning.

We have revised this sentence and integrated the statement into the previous ones. It now reads:

"Similar uplifting of Chl-a isolines towards the surface have been reported in other eddies (Lochte and Pfannkuche 1987; Feng et al., 2007; Noyon et al., 2019) and have been suggested to rusult from phytoplankton relocation through intense vertical mixing from strong surface winds (Feng et al., 2007; Noyon et al., 2019). Before our eddy survey, strong surface winds occurred offshore (SI Fig. S5), which might explain the high Chl-a concentration (>0.5 µg L-1) that we found at the surface (5 m) in all stations within the CE." Lines 569-574.

Lines 576-580: delete as this is mostly results.

We have deleted this statement.

Lines 594-598: again very speculative. The authors do not have data about the presence of diatoms or dinoflagellates in this study. Delete or rewrite.

We have deleted this statement.

Line 603: the use of the term "diversity" is not appropriate here, as the authors only provide data of a couple of functional phytoplankton groups.

We now avoid the term "diversity".

Lines 603-606: revise English usage as it is very difficult to understand the sentence.

We have revised the sentence. It now reads:

"Our flow cytometry data (SI Fig. S6) showed that Cyanobacteria (Synechococcus) and eukaryotic pico- and nanoplankton within the CE were unevenly distributed. This suggests that the phytoplankton community of the CE was likely distinct from the surrounding waters, but also variable on the submesoscale within the CE. This is consistent with previous studies on phytoplankton distributions in eddies (e.g., Lochte and Pfannkuche, 1987; Lasternas et al., 2013; Hernández-Hernández et al., 2020)." Lines 587-592.

Lines 607-608: delete this sentence, the authors do not have data on phytoplankton taxonomy, only flow cytometry counts of different groups based on scatter and fluorescence.

This sentence has been deleted.

Lines 619-621: this sentence is just repeating results. Please, delete.

This sentence has been deleted.

Line 621-625: please revise English usage.

The sentence has been revised. It now reads:

"We emit two hypotheses regarding this distribution: 1) the lower PER was due to a higher proportion of larger phytoplankton (e.g., diatoms), which have lower turnover rates and therefore lower PER and/or 2) the upwelling of nutrients generated by the CE might have enhanced the physiological health of the phytoplankton community." Lines 602-606.

Lines 630-631: please indicate where this correlation is found in the results, as the correlation matrix in figure 7 was calculated including all data.

We have divided original Figure 7 into two correlation matrices as suggested by the reviewer in an earlier comment and added the reference to the figure in the main text.

Lines 631-633: revise English usage. Else, it is hard to see such continuous trends in HB or PP in the figures.

We have changed the sentence to:

"Along the zonal transect, in the stations not affected by the eddy (open ocean+coastal stations), a significant positive correlation was observed between HB abundance and PPTOT rates (Fig. 7a)." Lines 609-611.

Line 635: delete this first sentence.

The sentence has been deleted.

Line 636: explain the acronym CanUS.

We now explain the acronym CanUS here.

Lines639-641: re-write for clarity. Again, avoid statements about phytoplankton compositions, as the authors are not reporting such data (they only have cytometric groups).

We have removed this statement.

Line 656: please town down, change "state" to "suggest".

We have made the change according to the referee's suggestion.

Lines 661-662: certainly BGEs are very low, which may be partially related to a severe underestimation of BP (see general comments and comments to the materials and methods section).

As outlined above, BP rates and BGEs have been corrected due to a mistake in the calculations.

Lines 686-688: please delete references tp the presence of diatoms and/or dinofñagellates as these data are not provided. Also town donw the statement.

We have changed the sentence to:

"As stated previously, the upwelling induced by the CE and the Frontal Zone led to higher phytoplankton biomass, which was likely responsible for this increase in BP." Lines 646-648.

Lines 689-706: all this discussion must be revised once BP estimates are clarified. Also engñish usage should be revised.

The paragraph has been revised after the correction as outlined above. Lines 646-656.

Lines 707-714: all this paragraph is about an spurious relationship. In addition, the authors do not have data on bacterial community compostion. I suggest deleting it.

We have deleted this part of the discussion according to the referee's suggestion.

Line 715: revise the usage of the term "growth" as this variable was not included in this study.

We have deleted this part of the discussion.

Lines 720-722: revise as it is very difficult to follow the reasoning, as phytoplankton taxonomic composition is not provided in this study.

We have deleted this part of the discussion.

Lines 732-736: revise English usage. In addition, revise statements about temporal dynamics, which does not seem to be adequately resolved in this survey.

We have changed the sentences to:

"Here we showed that both autotrophy and heterotrophy can occur within a single eddy. This urges the need for more high-resolution eddy studies in order to better estimate their impact on plankton metabolic activities and carbon cycling." Lines 666-668.

**Response to comments from reviewer #2**

General comments:

This study mainly investigated how cyclonic eddy (CE) affects heterotrophic bacterial activities in the surface waters of the eastern tropical North Atlantic by using measurements of various parameters related to the microbial activities. The measurements are valuable for understanding the effect of CE on the microbial activities. The study is interesting and suitable for the scope of this journal. However, there are several points which should be made clearer before publication. Please find below specific comments.

Major comments:

In this study, bacterial biomass production (BP) and community respiration (CR) rates are the most important parameters. Those rates depend on in situ temperature. However, BP and CR were estimated not at in situ temperature but at 14 â„ ƒ . The reason why the authors used 14 â„ ƒ as incubation temperature should be mentioned.

We thank the reviewer for the thorough review of our manuscript. We agree that incubating at in situ temperatures would have been favorable. As explained as response to a comment from reviewer #1 regarding the same criticism, we had to choose a different incubation temperature than in situ temperature for CR for technical reasons and time constraints. To obtain comparable results for BP and CR, we have used the same temperature for both rate measurements, i.e., 14 °C. The well-documented dependence of CR and BP rates on temperature (e.g., López-Urrutia and Morán, 2007; Yvon-Durocher et al., 2012) allowed us to correct for the difference between incubation and in situ temperature. The temperature correction is explained in detail in the Methods section and the SI.

There are several points that are not based on the clear evidences:

1)bacterial respiration rates are related to semi-labile (SL) dissolved organic carbon (DOC) concentration (lines 651-652), 2) microbes in the CE preferentially use SL-DOC (lines 696-697), 3) microbes do not grow in tandem with the increase in dissolved primary production (PP$_{DOC}$) but are related to the different requirement between BR and BP (lines 715-717), and 4) bacterial growth efficiency (BGE) varies depending on both BP via phytoplankton taxonomical composition and BR via the quantity and quality of the SL-DOC (lines 720-722). The statements 1), 2) and 3) are probably based on the results of correlations between relevant parameters (Fig. 7), while the statement of 4) is probably based on Table 2, Figs. 6a,b and 7. The results that each statement is based on are not clear at present. Please make the statements clearer by referring to proper results.

The reviewer is correct. The statements 1-3 are based on Fig. 7 (Fig.6 in the original manuscript), while statement 4 is based on Fig. 7 (Fig.6 in the original manuscript) and Table 2. Some of the statements have been changed in the revised version due to comments from reviewer #1, but where appropriate, we now refer to the Figure or Table for each of the statements.

Specific comments:

Line 271: How long scintillation vials are left open after addition of HCl should be described and proper reference should also be added here. I wonder if all dissolved inorganic carbon can be removed by the method or not.

The scintillation vials were left open for 14 hours after addition of HCL according to the method described in Steemann Nielsen (1952). We have added this information and the reference. The blanks showed no evidence of remaining inorganic carbon.

Figure 3: Adding the depth profiles of BGE and $PP_{DOC}$ is helpful for readers.

We have added the depth profiles to the Figures 3 and 4, respectively.

Lines 569 and 593: The authors mentioned high vertical mixing due to strong surface winds. Showing the strong surface wind data would be helpful for readers.

The wind data was shown in Supplementary Figure 6 of the original submission. We now refer to this Figure here.

Lines 608-609: Mixed layer depths should be added to Figures 2, 3, and 4 for easy readability.

We have considered the reviewer's suggestion. However, after including the mixed layer depth in the plots, we felt that the figures might be overloaded. Instead, we now show the mixed layer depth in a new Figure (3a, see below) and in Table S1.

[Figure]

Figure 3: Spatial distribution of the mixed layer depth (a) and chlorophyll *a* integrated over the upper 100 m depth (b) during M156.

Lines 630-631: Please clarify whether all the data of HB abundance and particulate primary production or a part of those data were used

The statement was based on the correlation of the parameters in the open ocean and coastal stations only (stations not affected by the eddy). According to a comment from reviewer #1, we have included a correlation analysis for the eddy-influenced stations and the stations not influenced by the eddy in two panels A and B (see new figure below) and removed the original correlation analysis. We have referred to the new Figure and changed the text accordingly to make it clearer that the statement is based on the correlations for the stations not affected by the eddy.

[Figure]

**(a)**

Open ocean
+ Coastal

[Figure]

**(b)**

Cyclonic eddy
+ Frontal zone

New Figure 7: Pearson correlation matrix of biochemical parameters, metabolic activities, and bacterial abundance in the upper 100 m in samples not influenced by the cyclonic eddy (i.e., coastal and open ocean stations) (a) and samples influenced by the cyclonic eddy (b). Statistical significance: '\*\*\*'< 0.001, '\*\*'< 0.01, '\*'< 0.05.

---

## Referee Report (RR1)

**Review of manuscript "Eddy enhanced primary production sustains heterotrophic microbial activities in the Eastern Tropical North Atlantic"**

The authors have made a great effort revising the original manuscript and have adequately addressed all my concerns. I only have a few minor revisions/suggestions.

Line 335 an elsewhere. The term nutricline is not correctly used here a several other parts of the manuscript. I suggest changing it to "nutrient isolines".

Line 384. Revise this maximum value for integrated Chl-a (and also in Table 1) as the value is not coherent with figure 3, where the maximum values in the map is 160 mg m-2

Line 524. Revise this fragment "coupled but differently" for clarity. I guess the authors mean that both variables were more coupled than in the stations outside the eddy (correlation coefficient is higher)

Lines 603-606. The authors could extend a bit more this discussion adding relevant references dealing with variation of PER along productivity gradients and/or in relation to phytoplankton size.

Line 651. Please consider revising the expression "makes sense" as it seems to colloquial.

Line 652. I suggest changing "favourable" to "utilizable" or "available".

Lines 655-656. Very low BGEs could be related to nutrient limitation. If bacteria have C available but not inorganic nutrients, the building of biomass (i.e BP) may be limited.

---

## Author Response (AR2)

**Review of manuscript "Eddy enhanced primary production sustains heterotrophic microbial activities in the Eastern Tropical North Atlantic"**

The authors have made a great effort revising the original manuscript and have adequately addressed all my concerns. I only have a few minor revisions/suggestions.

We thank the reviewer for further supporting our manuscript and the helpful comments. We have made the changes according to the reviewer's suggestions as outlined below.

During the revision of our manuscript, we noticed that the algorithm used to calculate the mixed layer depth based on the density criterion after Levitus, 1982 (the depth at which a change from the surface density of 0.125 kg m$^{-3}$ has occurred) resulted in a slight underestimation of the mixed layer depth in a few profiles from the open ocean (E1, S1, E2, S3) and coastal (E5) stations. We have changed the depth of the mixed layer in these profiles (five out of 24) by visually checking them and making any necessary corrections manually. We have changed the text, tables (SI Table n°1), and figures (Figures 3 and 6) accordingly. Those changes do not affect the interpretation of the results and all trends remained the same. Please see below for a comparison of the mixed layer depth before and after manual correction.

[Figure]

Line 335 an elsewhere. The term nutricline is not correctly used here a several other parts of the manuscript. I suggest changing it to "nutrient isolines".

We have made the correction according to the referee's suggestion.

Line 384. Revise this maximum value for integrated Chl-a (and also in Table 1) as the value is not coherent with figure 3, where the maximum values in the map is 160 mg m-2

We have revised the values and changed figure 3 accordingly.

Line 524. Revise this fragment "coupled but differently" for clarity. I guess the authors mean that both variables were more coupled than in the stations outside the eddy (correlation coefficient is higher)

We have changed the text to:

In contrast to the stations outside the eddy, HB was not correlated to PP$_{TOT}$, PP$_{DOC}$ and SL-DOC (p>0.05), but was strongly correlated to Chl-a and autotrophic pico-and nanoplankton biomass (r=0.57 and 0.76, respectively, p<0.001). Lines 523-525.

Lines 603-606. The authors could extend a bit more this discussion adding relevant references dealing with variation of PER along productivity gradients and/or in relation to phytoplankton size.

We have made the correction according to the referee's suggestion.

Line 651. Please consider revising the expression "makes sense" as it seems to colloquial.

We have changed this sentence to: "SL-DOC concentrations showed a strong positive correlation with BR, indicating that high molecular weight DOC compounds (>1 kDa) are an available carbon source for heterotrophic microbes (Amon and Benner, 1994, 1996; Benner and Amon, 2015)." Lines 650-653.

Line 652. I suggest changing "favourable" to "utilizable" or "available".

We have made the correction according to the referee's suggestion.

Lines 655-656. Very low BGEs could be related to nutrient limitation. If bacteria have C available but not inorganic nutrients, the building of biomass (i.e BP) may be limited.

We have added the following sentence to add this possibility: "One explanation might be that variability of nutrient availability in the surface waters limited the building of bacterial biomass (Thingstad et al., 1997; Janson et al., 2006; Berggren et al., 2010) but this requires further studies.". Lines 656-659.